# Exploring the Potential of Machine Learning for Modeling Growth Dynamics in an Uneven-Aged Forest at the Level of Diameter Classes: A Comparative Analysis of Two Modeling Approaches

**Gaspard Dumollard**

School of Agricultural, Forest and Food Sciences HAFL, Bern University of Applied Sciences BFH, 3052 Zollikofen, Switzerland; gaspard.dumollard@bfh.ch

**Abstract:** Growth models of uneven-aged forests on the diameter class level can support silvicultural decision making. Machine learning brings added value to the modeling of dynamics at the stand or individual tree level based on data from permanent plots. The objective of this study is to explore the potential of machine learning for modeling growth dynamics in uneven-aged forests at the diameter class level based on inventory data from practice. Two main modeling approaches are conducted and compared: (i) fine-tuned linear models differentiated per diameter class, (ii) an artificial neural network (multilayer perceptron) trained on all diameter classes. The models are trained on the inventory data of the Canton of Neuchâtel (Switzerland), which are area-wide data without individual tree-level growth monitoring. Both approaches produce convincing results for predicting future diameter distributions. The linear models perform better at the individual diameter class level with test $R^2$ typically between 50% and 70% for predicting increments in the numbers of stems at the diameter class level. From a methodological perspective, the multilayer perceptron implementation is much simpler than the fine-tuning of linear models. The linear models developed in this study achieve sufficient performance for practical decision support.

**Keywords:** uneven-aged forest management; forest growth modeling; machine learning; diameter distribution; silvicultural decision support

## 1. Introduction

Uneven-aged forests are generally defined as forests whose stands contain several cohorts of trees of different ages, which results in a relative structural heterogeneity of these stands. Despite the diversity of uneven-aged management methods, the management of these stands is often associated with the maintenance of a targeted distribution of diameters [1]. A good understanding of the dynamics within an uneven-aged stand at the global level but also at the level of diameter classes then allows for better planning of cuts, in the sense of choosing the timing and the intensity of the next cut [2], including the intensity of the cut by diameter class. Growth models for uneven-aged stands can therefore support silvicultural decision making.

The literature on existing uneven-aged forest growth models identifies three possible resolutions for this type of model: the whole stand, diameter classes and individual trees [3,4]. It is also possible to make a distinction between empirical and mechanistic (also called process-based) models [3]. This distinction is theoretically relevant, although there is a gradient between these two archetypes, with models estimated on the basis of empirical data but whose structure is to some extent inspired by theoretical mechanisms of forest growth [5].

Stand-level models aim at predicting the increment of a key stand-level variable, typically the increment of the standing volume or basal area. The recent literature has seen the publication of several such studies based on machine learning methods and data from permanent plots [6–8]. In particular, some of these studies use artificial neural networks among other regressors [7,8]. All of these models achieve good predictive performance and can be useful in practice, e.g., in management systems where harvested volumes are planned on the basis of increments. However, they do not provide information on stand structure.

Models at the individual tree level aim to predict individual diameter increments. Several recent studies have approached this topic from a machine learning perspective [4,9], for example, by testing several types of models including an artificial neural network [9]. These models perform well and give a maximum of details on growth within a stand, but individual monitoring data on diameter growth, i.e., data from permanent plots, are needed to train them. When this is the case, the data are usually available in large numbers, which fits well to machine learning approaches. For example, one of these studies [4] is based on 16,619 observations from the monitoring of 20 permanent plots.

Models based on diameter classes generally aim to predict increments in the number of stems per diameter class. Some studies [5,10] propose for example an empirical modeling of radial increments per diameter class based on inventory data using a linear model. These radial increments are then integrated into a process-based model to simulate the dynamics of the passage of trees between two successive classes, as well as the recruitment into the first diameter class. Numerous other studies [11–14] use an approach that is quite similar in principle but generalized and formalized. Passage rates are modeled directly (and empirically with linear models), and the increment in the number of stems in a given diameter class potentially depends on the respective numbers of stems in all other diameter classes. This type of model accounts for passages between two non-successive classes, competition effects between diameter classes, as well as the influence of the diameter distribution on recruitment. These models are formalized as matrix equation systems allowing the recursive simulation of stand dynamics, sometimes called a matrix model or transition matrix model [3]. The predictive performances of those models are limited compared to recent machine learning models developed on the stand or individual tree levels.

The canton of Neuchâtel in Switzerland is historically and professionally recognized as the Swiss high place of the selective felling forest management, which is implemented on a tree-by-tree basis (see Section 2.2 for further details). In the canton of Neuchâtel, selective felling management is implemented through the so-called control method (*Méthode du contrôle*) [3,15,16]. This method is based on periodic and complete inventories of stands as well as on a direct control of all harvests. Some studies [5,10] use an extract of these data. Some other studies [12,14] use data from the French Jura mountain where a similar management and controlling system prevail. These inventory data from the practice are non-experimental and are not based on individual tree monitoring, which makes their analysis more complex than data from permanent plots.

The objective of this study is to explore the potential of machine learning for modeling growth dynamics at the level of diameter classes from the available inventory data of the Canton of Neuchâtel. From a practical point of view, the aim is to determine if a model of this type can predict the evolution of the distribution of diameters that is specific to a given stand with sufficient accuracy to serve as decision support for the planning and implementation of cuts.

To achieve this goal, a machine learning workflow is implemented for data preparation and preprocessing, modeling, and evaluation of the models for their out-of-the-box predictive performances. This workflow can iteratively refine the data preprocessing and modeling steps. As to the modeling step, two main approaches are compared: (i) an approach based on fine-tuned linear models differentiated per diameter class, analogous in spirit to previous matrix models [11], (ii) an approach based on an artificial neural net-

work trained on all diameter classes. The potential of these two types of models for predicting a future state at the stand level is analyzed in absolute terms, but they are also compared with each other and with previous models found in the literature. Moreover, a comparative analysis of features importance and the impact of data scarcity on model performances is proposed. The best model is also evaluated for its ability to be used in practice, i.e., for its ability to predict the evolution of the overall diameter distribution at the stand level, and to predict increments of important aggregated variables.

## 2. Materials and Methods

### 2.1. Geographical and Climatic Context in the Canton of Neuchâtel

The canton of Neuchâtel, in Switzerland, is located in the Jura Mountain range. Its altitude stretches between 429 and 1552 m. Its climate is humid continental [17]. In terms of altitudinal zonation, the territory of the canton is currently located in the submontane, lower montane and upper montane zones [18]. Simulations of two possible climatic futures [18] also tend to show, in the canton, a transition from the lower and upper montane zones to the submontane zone. The southern part of the Canton, located at a lower altitude and in the minority, would even enter the foothill zone.

### 2.2. Brief Summary on Forest Management and Forest Data Collection in the Canton of Neuchâtel

The selection felling management that is historically applied in the canton of Neuchâtel is an uneven-aged management method where silviculture is implemented on a tree-by-tree basis. This type of management is based on periodic and situative cuts. The selection of trees to be harvested is the result of multiple determinants (wood harvesting, fostering tree vitality, fostering wood quality, continuous regeneration), but these cuts affect all diameter classes and are characterized by a fairly constant harvesting intensity that depends on the periodicity of interventions [2]. At the stand level, selective felling management results in the transformation to or the maintenance of a state of equilibrium that can be expressed in terms of diameter distribution within a given stand. A typical example of a diameter distribution at the stand level is shown in Figure 1.

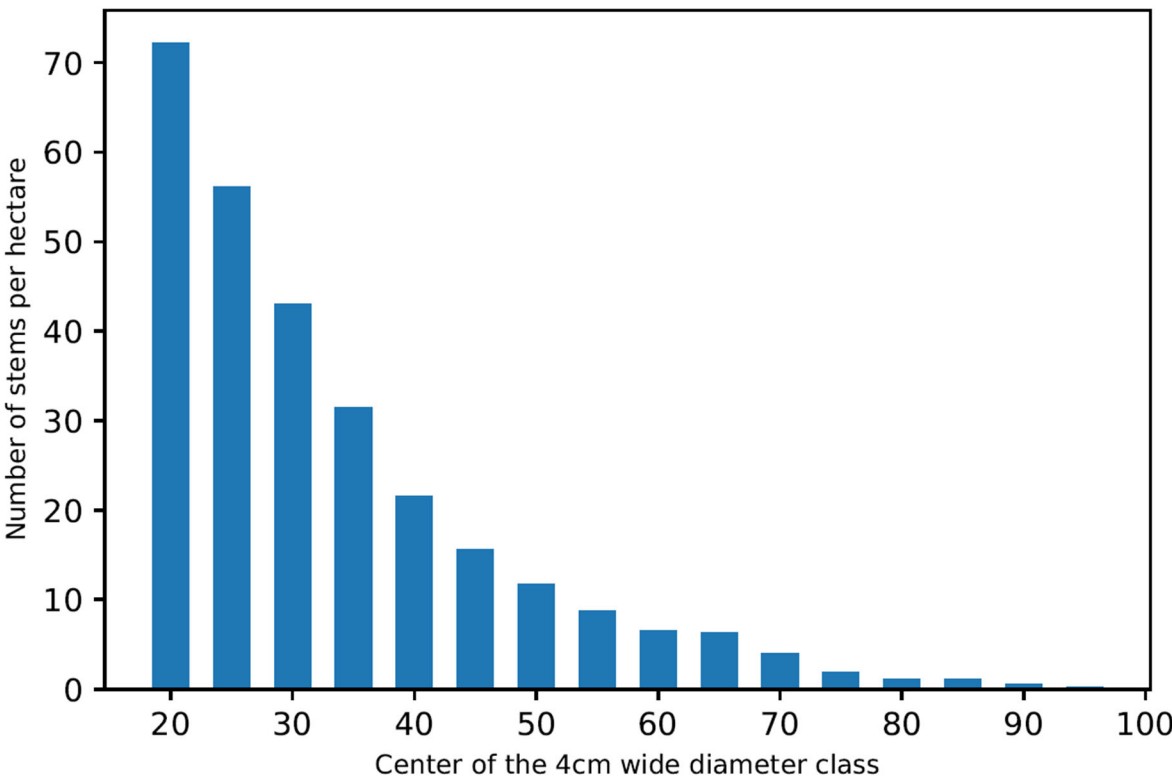

**Figure 1.** Example of a distribution of diameters in a forest division under selective felling management (division number 2202, inventory of 1971).

Information on the state of a stand, including the diameter distribution, also helps with the planning of the next cuts, as management goals on the stand level can be expressed as a desired state of equilibrium in terms of species mixture, standing volume, and stand structure [2].

Cuts take place periodically, typically every 7 to 12 years (in the Canton of Neuchâtel). The maintenance of the equilibrium is made possible by the fact that the harvesting of given trees allows for a "continuous" natural regeneration within the stand and allows for the further growth of trees in the intermediate strata.

Management by selective felling on a tree-by-tree basis is generally based on species that are shade tolerant, at least in their youth, typically but not exclusively white fir, spruce and beech. Since regeneration occurs in small patches and trees continue to grow in intermediate strata in the shade of larger trees, heliophilic species are not suitable for this type of management.

In the canton of Neuchâtel, selective felling management has been implemented through the so-called control method (*Méthode du contrôle*) [3,15,16] since the end of the 19th century. This method is based on several important elements: complete inventory of the stands (in particular individual diameters and species) between two successive cuts, direct control (i.e., inventory) of all harvests (as well as forced harvests), definition of parcels with fixed boundaries as management units called divisions (a term that we use hereafter in lieu of stand). The diameters are recorded by class of 5 cm width and with an inventory threshold of 17.5 cm. The classes are numbered from 1 to 24, and the center of the class in cm is then obtained by the following formula ((#class − 1) * 5 + 20). From the perspective of selective felling management, the purpose of collecting such systematic and detailed information is to be able to accurately characterize the silvicultural state of a division, particularly with respect to the diameter distribution, so as to better plan the next cuts. This information also allows for a detailed monitoring of forests at the division or landscape level over time and thus helps in better forest planning at the landscape level.

There are no direct data on recruitment and natural mortality (excluding windfall cases) that are collected through this control method. The application of this method in Neuchâtel allowed for the collection of the large data set used in this study.

*2.3. Data and Data Preprocessing*

This study is based on a standard machine learning workflow. The data collection and preprocessing steps are presented in this section. Section 2.4 presents the data sets building and features engineering steps, and Section 2.5 presents the modeling approaches.

The raw data consist of a record of the absolute number of stems in a division for a given inventory year, species and diameter class. These data are subjected to preprocessing, which consists of the following steps.

Step 1. The absolute numbers of stems per division are converted into numbers of stems per hectare. The areas of the divisions, whose boundaries are administratively set and (persistently) fixed, are known and could be found for 98% of the divisions mentioned in the data set. The remaining divisions were discarded. Meanwhile, the main forest site of each division (in terms of area) is integrated into the data set. Forest sites are given according to the typology of Ellenberg and Klötzli [19].

Step 2. Only divisions that have been inventoried at least twice are kept, i.e., divisions for which it is possible to calculate increments over at least one period are kept, which corresponds to 95% of the divisions kept thus far. From this step on, the inventory data are organized by couples (division, inventory year). This couple designates the state of a division at a given date, but it also designates the period between the inventory year and the date of the next inventory to which cuts and windfalls are associated (see step 4) and for which it is possible to determine increments.

Step 3. Only the couples (division, inventory year) whose species are compatible with selective felling management are retained. The three usual species in this management method are white fir, spruce and beech [16], but the presence of other species that are sufficiently tolerant to shade, in the minority or minority, cannot be excluded. Therefore, the following criteria were defined: the majority species (in terms of basal area) must be among fir, spruce or beech, the 2nd majority species must be among fir, spruce, beech, maple, or ash, and the 3rd majority species must be among the same species or flowering ash or be another deciduous species.

Table 1 is a contingency table of the number of couples (division, inventory year) according to their first and second majority species, before application of step 3. The table reveals that the divisions that meet the filtration criteria of step 3 are overwhelmingly in the majority, reflecting the importance of the selective felling management in the Canton of Neuchâtel. The table also reveals that most of the situations that do not meet the defined criteria correspond to couples (division, inventory year) where the 1st or 2nd majority species is oak. These situations may correspond to even-aged stands preferentially located in the south of the canton on less elevated grounds and on forest sites that are better adapted to oak.

**Table 1.** Contingency table of couples (division, inventory year) according to the 1st and 2nd majority species (for simplicity of representation, the 2nd majority species that had less than 10 couples for all 1st majority species combined are not reported in this table).

| | | 2nd Majority Species | | | | | | | | |
|---|---|---|---|---|---|---|---|---|---|---|
| | | Spruce | Fir | Beech | Oak (Sessile) | Scots Pine | Maple (Sycamore) | Ash | Other Deciduous | Aspen | Other Resinous |
| 1st majority species | Fir | 1412 | 0 | 544 | 5 | 5 | 3 | 1 | 11 | 15 | 0 |
| | Spruce | 0 | 1053 | 444 | 15 | 6 | 85 | 7 | 23 | 8 | 0 |
| | Beech | 201 | 317 | 0 | 143 | 19 | 25 | 5 | 2 | 0 | 2 |

| Oak (sessile) | 2 | 14 | 124 | 0 | 5 | 0 | 0 | 0 | 0 | 8 |
| Scots pine | 5 | 16 | 18 | 7 | 0 | 0 | 0 | 2 | 0 | 0 |
| Larch | 1 | 0 | 0 | 0 | 0 | 0 | 0 | 0 | 0 | 0 |
| Other resinous | 1 | 0 | 14 | 9 | 0 | 0 | 0 | 4 | 0 | 0 |
| Other deciduous | 3 | 12 | 0 | 0 | 3 | 0 | 0 | 0 | 0 | 0 |
| Black pine | 3 | 0 | 0 | 3 | 2 | 0 | 0 | 0 | 0 | 0 |
| Poplar | 0 | 0 | 0 | 0 | 1 | 0 | 2 | 0 | 0 | 0 |
| Ash | 4 | 0 | 0 | 1 | 0 | 0 | 0 | 0 | 0 | 0 |

Step 4. The inventory data are merged with the respective data on decided cuts and windfalls, which are formatted in the same way. The data on cuts and windfalls are associated with couples (division, inventory year) when the cuts and windfalls occur in this division and between the inventory year and the date of the next inventory.

Step 5. The temporal gap between two successive inventories in the same division is variable, see Figure 2.

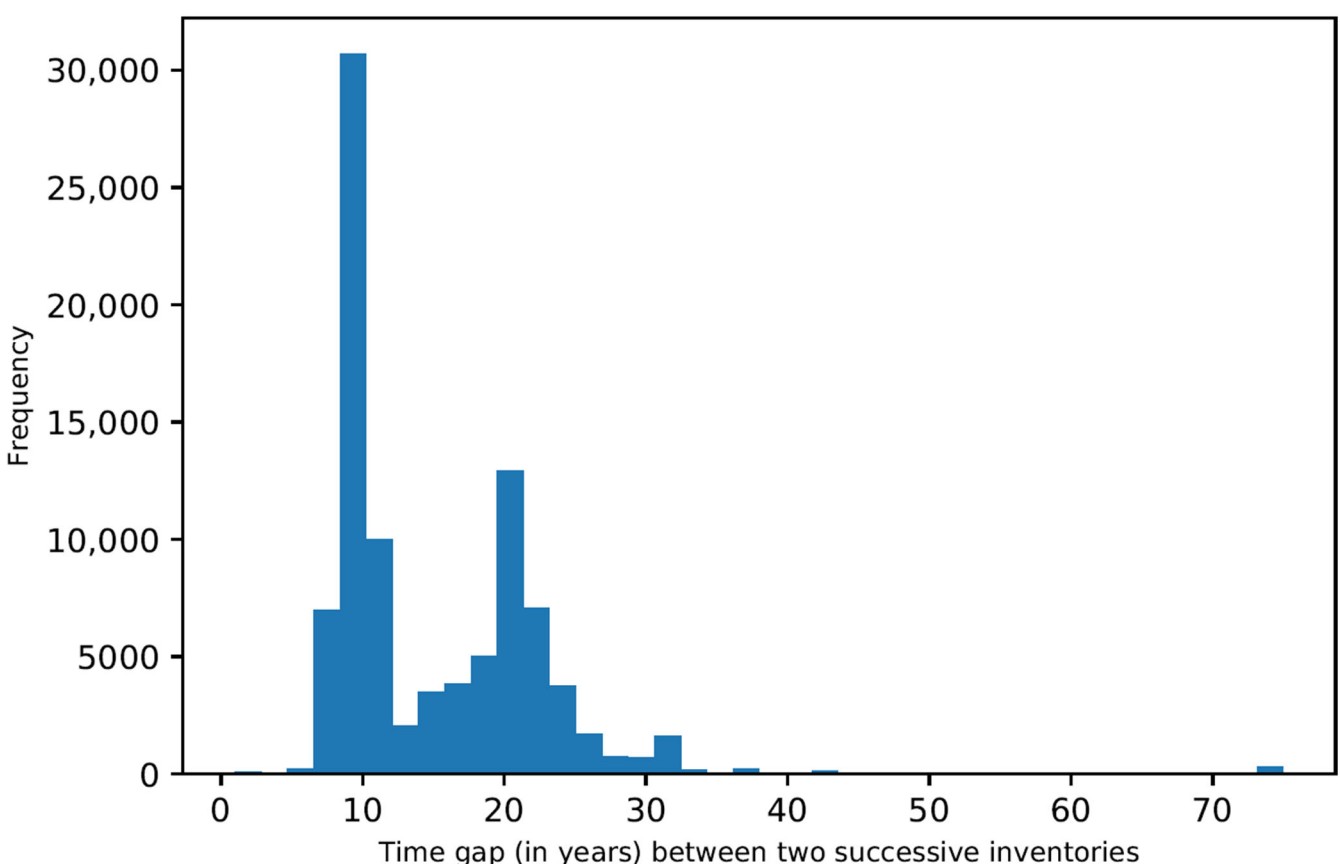

**Figure 2.** Histogram of the time gap between two successive inventories in a given division.

There are a large number of successive inventories that are 7 to 12 years apart but also many successive inventories that are 20 to 24 years apart. In the first case, the inventories were carried out before and after the same cut and correspond therefore to the standard application of the control method. In the second case, there are several cuts between two successive inventories. The standard application of the control method is actually historical in the Canton of Neuchâtel, but it has since been adapted, and the current

legislation provides for an inventory at least every 25 years for each division, which corresponds to 2 or 3 cutting cycles. In some divisions of historical or technical importance, inventories are maintained on the basis of one inventory per cutting cycle.

The goal of this study is to simulate growth dynamics recursively, i.e., we want to predict the state of a division at date $t + \Delta t$ from its state at date $t$. This is a pragmatic approach that is well suited to the development of the model and its use in practice. The duration $\Delta t$ does not need to be constant (see Section 3), but it must be within an interval whose definition meets two constraints: (i) the possibility of finding data describing dynamics over such periods; (ii) the relevance of these durations $\Delta t$ for a use of the model as a decision support in practice. Figure 2 shows that the available data correspond to durations from 8 to 26 years. However, this interval is too wide for model development. The diameter data are given in 5 cm diameter classes. For a relatively large annual diameter increase of 5 mm.year$^{-1}$, a tree takes 10 years to completely cross a diameter class. Over a period of 20 years and more, with such an increase, a tree can pass through two or more diameter classes. The growth mechanisms in the sense of recursive modeling cannot therefore be the same over these two types of periods. We choose to focus on durations $\Delta t$ of up to 12 years. This interval allows the model to be used over a sufficiently long period to assist in planning the next cut, and it is possible to use the models recursively to simulate dynamics over longer periods. Data corresponding to gaps of more than 12 years between two successive inventories are left out, so that only 52% of couples (division, inventory year) that had been retained until then are kept.

Step 6. Much of the harvest data are not available. All the data on the cuts, some of which are old, have not yet been compiled and digitized. This gap in the data leads us to keep only the couples (divisions, year of inventory) for which cuts and windfalls are known, i.e., 38% of the couples that had been kept until now. In addition, a few outliers (in small numbers) that stand out by cutting intensities that are much too strong for the selective felling management are also set aside.

Step 7. The number of trees in the upper diameter classes is small or even negligible and therefore does not allow for a reliable modeling of the growth mechanisms among these classes. Therefore, the data must be truncated at a given diameter threshold. The number of trees that are beyond diameter class 16 (class 24 is the maximum class), i.e., trees with a diameter strictly greater than 97.5 cm, are relatively rare compared to the lower classes; they represent about 0.07% of the trees remaining in the data set at this stage. Such a threshold is also high enough that it is not a problem for practical use of the model; thus, we chose this threshold to truncate the data. Nevertheless, simply ignoring trees beyond class 16 could be problematic in some cases. These trees are few in number but large and can therefore have a non-negligible impact on the growth of other trees. For this reason, only couples (division, inventory year) that meet the following criteria are retained: a maximum of 0.5 trees per ha above class 16 in the division and a maximum of 0.125 trees per ha above class 16 for cuts or windfalls. These criteria were defined partly arbitrarily but in the search for a compromise between filtering potentially biased data and keeping a sufficient volume of data. At the end of this step, 77% of the couples (division, year of inventory) that had been kept until then remain.

In the end, after this preprocessing, there are data on 580 couples (division, inventory year), and for each couple, there are data on 16 diameter classes. Individual observations in the data set then correspond to triplets (division, inventory year, diameter class). The mean basal area of remaining couples (division, inventory year) is 28.0 m$^2$.ha$^{-1}$ (the standard deviation is 7.26). Despite a strong filtration, the volume of data remains important for a study of this type. The data are moreover complete and consistent.

*2.4. Data Buildings and Features Engineering*

The objective is to develop a model or a set of models to predict the gross annual increment in the number of stems per hectare for a diameter class $d$ (if there were neither

cuts nor windfalls, hence the gross increments), noted $\Delta N_d$. $\Delta N_d$ will be the endogenous variables in our models. They can be calculated as follows:

$$\Delta N_d = \frac{(N_{after,d} - N_d + N_{cut,d} + N_{windfall,d})}{\Delta t} \tag{1}$$

In Equation (1), $N_d$ denotes the number of stems per hectare in diameter class $d$ at the time of the first inventory, $N_{after,d}$ the number of stems per hectare in diameter class $d$ at the time of the subsequent inventory, $N_{cut,d}$ and $N_{windfall,d}$ the respective numbers of stems cut or windfallen per hectare in diameter class $d$ between these two inventories, and $\Delta t$ denotes the duration in years between the two successive inventories.

Following exogenous variables (also called features) are defined for use in the models (all of which are expressed per hectare, except the diameter class):

1.  The diameter class $d$;
2.  The number of stems per diameter class at the time of the first inventory, aggregated by species type (resinous/deciduous) $N_{resinous,d'}$ and $N_{deciduous,d'}$, or aggregated for all species $N_{d'}$ (for $1 \le d' \le 16$);
3.  The corresponding basal areas (calculated from the $N$ features and the centers of diameter classes): $G_{resinous,d'}$, $G_{deciduous,d'}$, and $G_{d'}$ (for $1 \le d' \le 16$);
4.  The numbers of stems cut and windfallen (aggregated for all species) and expressed on an annual basis:

$$N_{annual\ cut,d} = \frac{N_{cut,d}}{\Delta t}, \tag{2}$$

$$N_{annual\ windfall,d} = \frac{N_{windfall,d}}{\Delta t}. \tag{3}$$

- The total number of stems per hectare $N_{tot}$ and the total basal area per hectare $G_{tot}$ are calculated too;
- The basal area of all trees overlying a given diameter class is used as an indicator of the competition to which this diameter class is subject. This variable, denoted $G_{cum}$ (for cumulative basal area), is defined as the sum of the basal areas of every tree larger than the considered diameter class [4,5], in our case:

$$G_{cum.d} = \sum_{d'=d+1}^{d_{max}} G_{d'}. \tag{4}$$

- The number of stems cut or windfallen annually in the overlying classes (as an indicator of the regulation of competition for a given diameter class):

$$N_{overlying\ annual\ cut,d} = \sum_{d'=d+1}^{16} N_{annual\ cut,d'}, \tag{5}$$

$$N_{overlying\ annual\ windfall,d} = \sum_{d'=d+1}^{16} N_{annual\ windfall,d'}. \tag{6}$$

This set of features includes indicators of the current state and structure of the division ($N_{resinous,d'}$, $N_{deciduous,d'}$, $N_d$, $G_{resinous,d'}$, $G_{deciduous,d'}$, $G_{d'}$), indicators of density at the division level ($N_{tot}$, $G_{tot}$), an indicator of competition at the diameter class level ($G_{cum.d}$), and indicators of sylvicultural management ($N_{annual\ cut,d}$, $N_{annual\ windfall,d}$, $N_{overlying\ annual\ cut,d}$, $N_{overlying\ annual\ windfall,d}$), all of which can potentially provide additional insight into forest growth. As an additional explanation, the $G_{cum}$ of a diameter class measures the density of trees in overlying diameter classes. Since competition between trees is on average exerted by larger trees on smaller trees, $G_{cum}$ effectively measures the competition exerted on a given diameter class within a given division. When some of the trees in these overlying diameter classes are cut or windfallen, the competition exerted on the diameter class may decrease, hence the role of $N_{overlying\ annual\ cut}$ and $N_{overlying\ annual\ windfall}$.

These dendrometric variables represent a classical set of variables used in forest sciences to describe the state and dynamics of a forest stand. In addition to these dendrometric variables, the main forest site for each division is also used as a feature [19].

*2.5. Modeling Approaches*

This study is based on two main modeling approaches: (i) fine-tuned linear models; (ii) an artificial neural network (a multilayer perceptron regressor). The criterion for refining and comparing models is a coefficient of determination $R^2$ calculated on a set of test data that are not used for training the models. This test data set accounts for 20% of total available data. Test data are selected among couples (division, inventory year), not at the level of triplets (division, inventory year, diameter class). Prediction performances are therefore measured on divisions that are completely unknown to the models, which is crucial to test the capacity of the model under practical conditions. The principle of parsimony, which with equal or almost equal performance, leads to favoring the simplest models, is also applied, as well as the search for models that are consistent with theoretical considerations on forest growth.

Algorithms for data transformation, model training and model evaluation are derived from the sklearn library (version 0.23.1, and version 1.0.2 for the SplineTransformer, [20]). Other important libraries are pandas (version 0.24.2, [21]) and numpy (version 1.19.5, [22]).

2.5.1. Detailed Method for Linear Modeling

Linear models are refined following a usual machine learning workflow so as to find the best possible combination of features, data transformation and (linear) regressor. This type of model provides transparent and easily interpretable results from the point of view of forest growth. Moreover, their linearity and the relatively limited number of parameters make them resilient to overfitting problems when sufficient data are available.

Past studies clearly show the essential role of diameter as a predictor of diameter growth [5,9]. However, the effect of diameter is not necessarily separable from the effect of other features. For example, there is no reason that the effect of forest density, competition between trees or harvesting be expressed in the same way on trees of (very) different diameters. In order to take into account these interaction effects (which could turn out to be non-linear) without complicating the modeling process, a distinct model is estimated for each diameter class, resulting in a total of 16 linear models. In order to maintain overall consistency, a common specification (choice of features) of these different models is proposed; only the transformation of the features and the estimation of the coefficients are distinct.

The selection of features is an important and sensitive step in a machine learning or statistical workflow. The principle of parsimony suggests keeping only the features that bring significant additional information to the model. This allows one to obtain a more robust model by limiting multicollinearity problems and facilitates the interpretation of the regression results.

In this study, the features selection is based on a mix of three approaches: (i) Recursive Feature Elimination; (ii) Stepwise Regression by Backward Elimination; (iii) theoretical considerations on forest growth.

Recursive Feature Elimination consists in successively and iteratively removing features from a model from the least to the most important one [23,24]. In the context of a linear model, the relative importance of the features is given by the numerical order of the regression coefficients in terms of absolute value (which are recalculated at each iteration). This purely empirical method makes it possible to obtain a first systematic ranking of the features according to the information they bring to the model. However, this approach is limiting for two reasons. The application of this method is sensitive to scale effects and therefore to the transformation of features (e.g., standardization). Furthermore, this approach does not take into account multicollinearity effects.

In order to complete this approach, a Stepwise Regression method with Backward Elimination is applied [25]. This method is based on the successive and iterative elimination of features, but the elimination is performed on the basis of a model fit criterion. At each iteration, the eliminated features are those that decrease the least (increase the most) fit criterion, here a test $R^2$. This allows one to take into account the multicollinearity effects. In this study, this method is applied at the level of groups of features (e.g., forest sites, density indicators) so as to identify the groups of features that can be removed and those that should be kept in the model. Theoretical considerations help in the definition of groups and further engineering of features. Proceeding by groups also facilitates a common specification for all models.

The models are also optimized from the point of view of the transformation of the features (e.g., standardization) and the choice of a regressor (choice of the loss function, regularization, fine-tuning of hyperparameters). To do so, different specifications are tested in order to find the combination that maximizes the test $R^2$.

In all cases (features selection, estimation of the final models), the fit criterion used is a coefficient of determination ($R^2$) measured on a set of test data not used for training the model and representing 20% of available observations. The root mean squared error (RMSE) is also calculated as additional information (also computed on test data sets). The advantage of the RMSE is that it expresses an average estimation error in absolute value, in the unit of the predicted variable. The evaluations of test $R^2$ and RMSE are always averaged on 10 different train-test splits.

### 2.5.2. Detailed Method with Multilayer Perceptrons

The use of a multilayer perceptron [26,27], which is a nonlinear model, allows for a more generic and flexible modeling approach. A multilayer perceptron with at least one hidden layer (hence multilayer) can serve as a universal approximator [28], in the sense that it can approximate with a finite number of neurons any continuous function (on particular subdomains, see [28] for further details). The scikit-learn library [27] provides such a ready-to-use regressor. In this case, potential exogenous variables can be all pooled together because this model fits to non-linear effects and interaction effects. However, this type of model, usually based on a large number of parameters (at least larger than linear models), requires large volumes of data to avoid overfitting. In addition, results (the estimated values of parameters) are harder to interpret. The volume of data used in this study is comparatively large, but it is not large in the sense of big data. One of the objectives of this second modeling approach is to analyze the potential of the multilayer perceptron for this modeling problem based on a large but probably limiting data set.

The multilayer perceptron regressor was trained to directly predict the gross numbers of stems (without the removal effect from cuts or windfalls) by diameter class at the end of a growth period of duration $\Delta t$: $N_{after\ gross,d}$. This direct approach, without going through increments, resulted in better predictive performances. Only one model was trained instead of separate models for each diameter class. Therefore, the training of this model is based on 9280 entries, which is a comparatively large volume of data for this type of application.

The exogenous variables (features) presented in Section 2.4 were all retained, and several features were even added: the diameter (center of the class), the length of the growth period $\Delta t$, which differs from one couple (division, inventory year) to another, as well as the forest site. The model hyperparameters: the number of hidden layers, the number of neurons in hidden layers, and the regularization parameter [27], are tuned by gridsearch and cross-validation. The fit criterion is also a coefficient of determination ($R^2$) measured on test data representing 20% of available observations. The root mean squared error (RMSE) is computed too. Results are all averaged over 10 different train-test-splits.

A permutation importance procedure [29] is also applied to a fitted multilayer perceptron. This procedure determines the importance of each feature to the model by measuring the model score loss (here the test R²) when the entries for this feature are randomly shuffled (the shuffling and the measure are here repeated 30 times for each feature).

## 3. Results

### 3.1. Results of the Linear Modeling

3.1.1. Features Selection

The best results with linear models are obtained by training a separate model for each diameter class. These models are of the following form for any diameter class $1 \leq d \leq 16$ and for each couple (division, inventory year) $i$:

$$\Delta N''_{d,i} = \beta^d_0 + \beta^d_{d-2} \cdot N'_{d-2,i} + \beta^d_{d-1} \cdot N'_{d-1,i} + \beta^d_d \cdot N'_{d,i} + \beta^d_{d+1} \cdot N'_{d+1,i} + \cdots + \beta^d_{d_{\max(d)}}$$
$$\cdot N'_{d_{\max(d),i}} + \beta^d_{cut} \cdot N'_{annual\ cut,d,i} + \beta^d_{windfall} \cdot N'_{annual\ windfall,d,i} + \varepsilon_{d.i}, \quad (7)$$

$\beta^d_j$ are the regression coefficients corresponding to the model of diameter class $d$.

The results of the Recursive Feature Elimination procedure are presented in Table A1 in Appendix A (note that these results are based on the optimal features transformation and the optimal regressor presented in Section 3.1.2). The results show that the most important features for predicting the variation of the number of stems in a given diameter class $d$, i.e., $\Delta N_d$, are the initial number of stems $N_d$ and basal area $G_d$ in this class, as well as annual cuts $N_{annual\ cut,d}$ and to a lesser extent annual windfalls $N_{annual\ windfall,d}$ occurring in this class. The number of stems ($N_{resinous/deciduous,d'}$) and basal area ($G_{resinous/deciduous,d'}$) in the different classes ($1 \leq d' \leq 16$) play a highly variable role that differs in part between models. Annual cuts in the overlying diameter classes $N_{overlying\ annual\ cut,d}$ are of intermediate importance. The features $G_{cum}$, $G_{tot}$, $N_{overlying\ annual\ windfall,d}$, $N_{tot}$ and forest sites play a less important role.

Following this order of importance, a Stepwise Regression with Backward Elimination is conducted by a group of features. The detailed results are presented in Table A2 in Appendix A (note that these results are based on the optimal features transformation and the optimal regressor presented in Section 3.1.2). In this procedure, the features corresponding to the forest sites are eliminated, then $N_{tot}$ and $G_{tot}$, then $G_{cum,d}$, then $N_{overlying\ annual\ cut,d}$ and $N_{overlying\ annual\ windfall,d}$. Then, there is the problem of multicollinearity between the features $N_{resinous/deciduous,d}$ and $G_{resinous/deciduous,d}$ (due to the identity $G = N \cdot \pi \cdot \left(\frac{d}{2}\right)^2$). Since the objective of the models is to predict variations in the number of stems, the $G$ features are eliminated. Then, the $N_{resinous,d'}$ and $N_{deciduous,d'}$ features are added together, yielding the $N_{d'}$ features (with $1 \leq d' \leq 16$). Finally, the $N_{d'}$ features can be adjusted to take into account certain characteristics of forest growth, such as the fact that in the short/medium run (around 10 years), trees of a given diameter normally have little influence on the dynamics of much larger diameter classes (see a full explanation below in this section). The eliminations or adaptations mentioned thus far do not lead to a strong and uniform degradation of the test R² of the different models compared to the models with all features (see Table A2). However, the elimination of $N_{annual\ cut,d}$ and $N_{annual\ windfall,d}$ leads to a noticeable and uniform degradation of the models. Thus, these latter features are kept, resulting in the models given in Equation (7) above.

As to the adjustment of features $N_{d'}$, when modeling the increment in the number of stems for a diameter class $d$ ($\Delta N_d$), among underlying classes, only the two preceding ones ($d-1$ and $d-2$) provide relevant information. Some stems in these classes may enter class $d$ during the growth periods considered in the data, typically 7 to 12 years. Trees in classes lower than $d-2$ do not have the time to grow as much (at least 10 cm in diameter) over a period of 7 to 12 years. Moreover, trees in those lower classes normally exert no competition (or limited competition) on trees in class $d$. The corresponding variables can therefore be eliminated since they would only capture statistical noise (overfitting). This verifies empirically when fitting models on unadjusted features $N_{d'}$. In the

same way, it is not always necessary to keep all the variables corresponding to numbers of stems in classes higher than a given class, i.e., $d_{max}$ is not necessarily equal to 16 in all cases (see Equation (7)). These $d_{max}$ values are adjusted for each model separately so as to optimize the model's performances (through a trial and error approach). A detailed analysis of the value of the regression coefficients and the role of each feature in the linear models is presented in Section 3.1.4.

### 3.1.2. Features Transformation and Optimization of the Regression Procedure

Despite their relative simplicity, linear models can be significantly improved by transforming the endogenous and exogenous variables. The exogenous variables (features) are each subjected to a standard scaling (noted by an apostrophe in Equation (7) above). This usual transformation is obtained by removing the mean and scaling to unit variance. The endogenous variable is subjected to a Yeo-Johnson transformation [30]. This transformation is noted by a double apostrophe in Equation (7). This non-linear transformation can reduce the skewness of a distribution and solves the heteroscedasticity problem that was visible on the residuals plots. This transformation is similar to the Box-Cox transformation [31], which is better known, but it accepts negative values as an argument, which is necessary in our case. The Yeo-Johnson transformation improved the performance of the models of classes 1 to 12 (as a reminder, classes are 5 cm wide, and the center of a class in cm = (#class − 1) × 5 + 20). For classes 13 to 16, this transformation did not improve performances, and it even deteriorated them. The transformation was therefore applied only to classes 1 to 12. These two transformations (standard scaling of the exogenous variables and Yeo-Johnson transformation of the endogenous variable) were performed separately for each model. B-splines transformations [32,33] on the exogenous variables were also tested. This type of non-linear transformation, related to the generalized additive modeling (GAM) approach, allows for non-linearity effects between the respective exogenous variables and endogenous variable. The use of this transformation did not produce strong and unambiguous results (performances were slightly better for half of the diameter classes and slightly worse for half of the diameter classes with no particular order among these diameter classes). By principle of parsimony and to keep a common structure to our models, this transformation was left aside.

Finally, best performances were obtained using a Huber loss function [34,35]. This loss function is quadratic below a given error threshold but linear above this threshold, which can limit the weight of points that stand out (e.g., residual outliers). Moreover, a regularization term of type L2 is added to the loss function [36], which can control the overfitting in a systematic way. Note that this regularization term makes the models non-linear in the strict sense. A Stochastic Gradient Descent regressor [36] is used to train the model, and the hyperparameters related to the Huber loss function and the regularization term are tuned by gridsearch and cross-validation.

### 3.1.3. Performance of Linear Models

The performances obtained for the different linear models are presented in Table 2. As the train-test-split is random, it is repeated 10 times, and the table therefore presents averaged results. Table 2 gives the test $R^2$s for the prediction of transformed annual increments of the number of stems per hectare $\Delta N''_{predict,d,i}$. The table also shows the test $R^2$s for predicting gross (i.e., if neither cuts nor windfalls occur) stem counts at the end of the growth period, i.e., based on the errors between the $N_{after\ gross,d,i}$ and the $N_{predict\ after\ gross,d,i}$ [6]:

$$N_{after\ gross,d,i} = N_{after,d,i} + N_{cut,d,i} + N_{windfall,d,i}, \tag{8}$$

$$N_{predict\ after\ gross,d,i} = N_{d,i} + \Delta N_{predict,d,i} \cdot \Delta t. \tag{9}$$

**Table 2.** Performances of the models measured by R² and RMSE scores for respective prediction of $\Delta N''_{d,i}$ and $N_{after\ gross,d,i}$ (the overall R² is measured on the whole data set when the models are trained on the whole data set).

| Diameter Class | Diameter Range | Test R² for the Prediction of $\Delta N''_{d,i}$ | Test R² for the Prediction of $N_{after\ gross,d,i}$ | Overall R² for the Prediction of $N_{after\ gross,d,i}$ | Test RMSE for the Prediction of $N_{after\ gross,d,i}$ | Mean of Real $N_{after\ gross,d,i}$ Values |
|---|---|---|---|---|---|---|
| 1 | [ | 13.6% | 82.4% | 82.0% | 15.3 stems/ha | 66.2 stems/ha |
| 2 | [22.5;27.5[ | 55.2 | 93.5 | 93.2 | 7.1 | 49.0 |
| 3 | [27.5;32.5[ | 64.5 | 93.4 | 93.5 | 5.7 | 40.2 |
| 4 | [32.5;37.5[ | 71.6 | 92.9 | 92.4 | 4.4 | 32.2 |
| 5 | [37.5;42.5[ | 66.9 | 91.3 | 91.7 | 3.5 | 25.8 |
| 6 | [42.5;47.5[ | 60.5 | 89.2 | 91.4 | 2.9 | 19.8 |
| 7 | [47.5;52.5[ | 54.7 | 90.3 | 92.0 | 2.4 | 14.9 |
| 8 | [52.5;57.5[ | 49.0 | 88.7 | 90.8 | 2.1 | 10.4 |
| 9 | [57.5;62.5[ | 51.2 | 90.6 | 90.9 | 1.6 | 7.3 |
| 10 | [62.5;67.5[ | 56.1 | 90.4 | 90.9 | 1.4 | 4.7 |
| 11 | [67.5;72.5[ | 58.1 | 89.6 | 89.7 | 1.0 | 2.7 |
| 12 | [72.5;77.5[ | 61.5 | 88.0 | 88.0 | 0.7 | 1.6 |
| 13 | [77.5;82.5[ | 60.6 | 84.2 | 85.3 | 0.5 | 0.8 |
| 14 | [82.5;87.5[ | 50.5 | 74.5 | 79.7 | 0.4 | 0.4 |
| 15 | [87.5;92.5[ | 53.8 | 69.8 | 72.9 | 0.2 | 0.2 |
| 16 | [92.5;97.5[ | 57.3 | 65.9 | 70.8 | 0.1 | 0.1 |
| **All classes combined** | | | 90.3% | 96.7% | | |

The plots of the residuals (corresponding to one of the train-test-split) for the 16 models are presented in Figure A1 in Appendix B. A review of the histograms giving the distributions of these residuals (not shown here) shows Gaussian-type distributions centered in 0.

The results for the predictions of $\Delta N''_{d,i}$ are satisfying except for class 1 ($17.5cm \leq d < 22.5cm$). Since there are no data on the number of stems for diameter classes below 17.5 cm, it is impossible to precisely predict the number of trees that will enter class 1 during the growth period, i.e., the recruitment rate of our set of models, hence the limited performance. For the other models, test R²s range typically from 50% to 70%. These models manage to predict most of the variance of the $\Delta N''_{d,i}$ based on the initial state of a division $N'_{d,i}$ (and on cuts $N'_{annual\ cut,d,i}$ and windfalls $N'_{annual\ windfall,d,i}$) for cases that were not used for training the model. Using these predicted increments (after inverse transformation) to predict stem counts at the end of the growth period also produces satisfying results, mechanically better than the prediction of increments since the initial stem counts are known and largely explain the stem counts 7 to 12 years later. All diameter classes combined, the models predict future stem numbers with a test R² of 90.3%. The root mean squared error (RMSE) indicates the average difference between predicted values and real values in absolute terms. It is here computed on $N_{after\ gross,d,i}$ real and predicted values and can be compared to mean real values of $N_{after\ gross,d,i}$. Table 2 shows that RMSE values are satisfyingly small compared to mean values, although performances decrease noticeably for diameter classes above 10 ($d \geq 10$) and are mediocre for diameter classes above 13 ($d \geq 13$). These figures can be compared to the result obtained with the artificial neural network approach (see below).

The models are then retrained on the whole data set to determine the final regression coefficients. Table 2 presented above shows corresponding R²s, which are in this case training R² s. Those training R²s are close to the test R²s presented before, showing an

absence of overfitting (or limited in the higher diameter classes), which is normal given that the linear models have relatively few features and have been trained on a fairly large data set (580 observations each).

### 3.1.4. Analysis of the Regression Coefficients

The regression coefficients corresponding to the training on the whole data set are presented in Table A3 in Appendix B and are shown in Figure 3 as a heatmap for ease of exposition and interpretation. Because of the transformations undergone by the variables (Yeo-Johnson and standard scaling, for each diameter class separately), the coefficients are difficult to interpret in absolute terms, but their sign can be interpreted, as can their relative values within a given model.

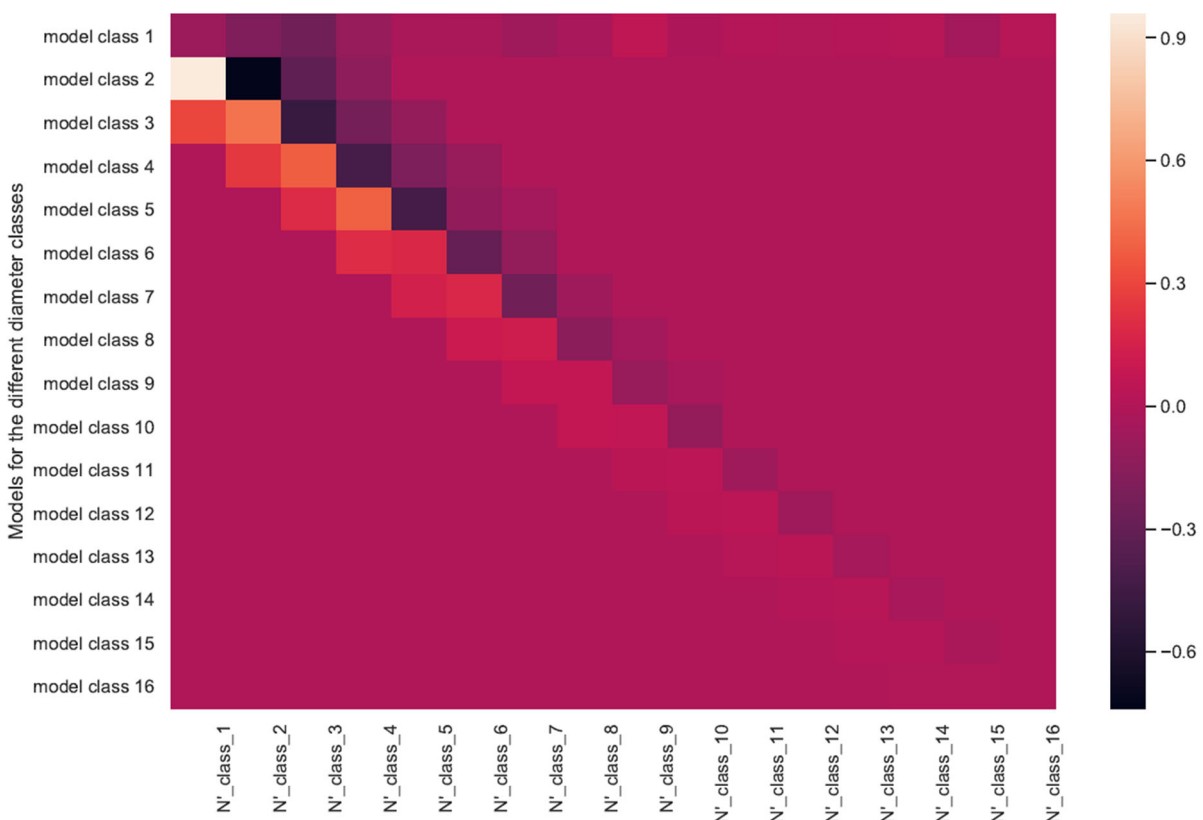

**Figure 3.** Heatmap of the regression coefficients for the models given by *Equation (7)* (the values are given in Table A3 in Appendix B).

For diameter classes greater than 1 (diameters greater than 22.5 cm), a common pattern is visible. This pattern is similar to the results presented in [13], despite the differences due to our data transformations. This pattern is also clearly visible when displaying the coefficients for a given model on a graph, see for example, Figures 4 and 5.

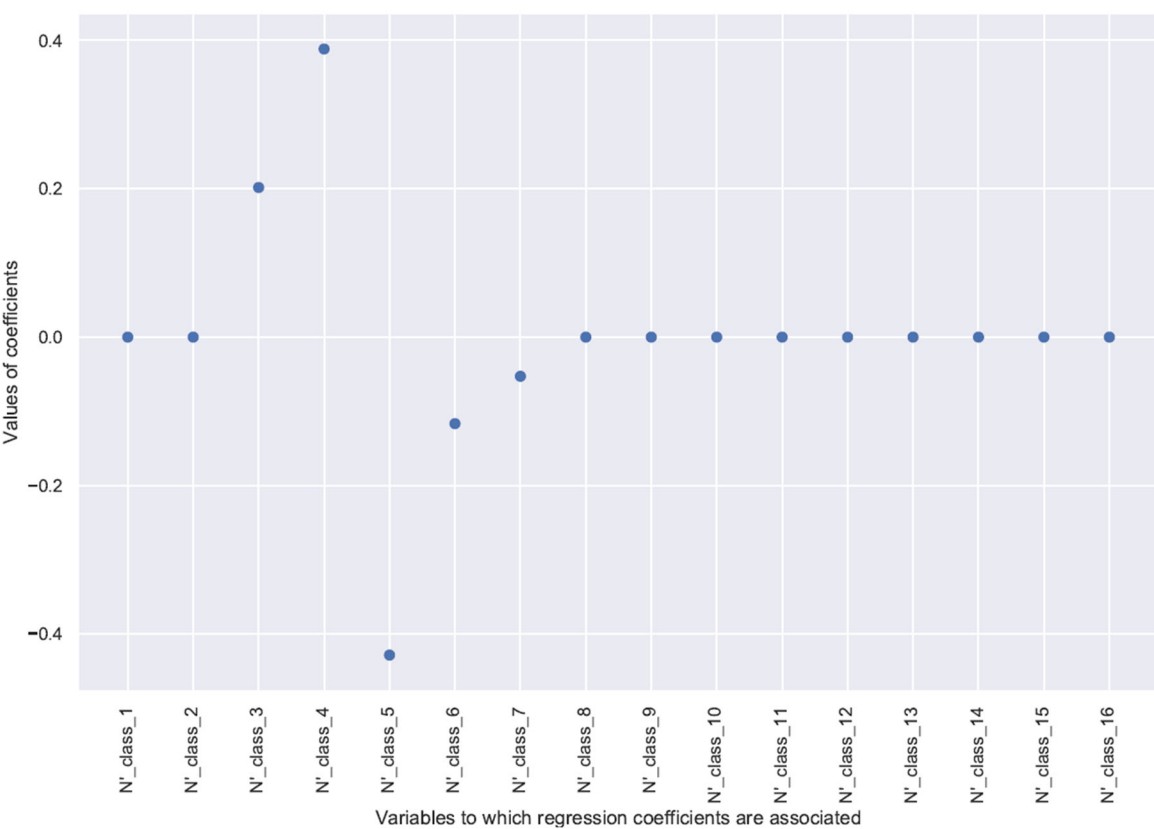

**Figure 4.** Regression coefficients for the model of diameter class 5, associated with transformed features $N_d'$.

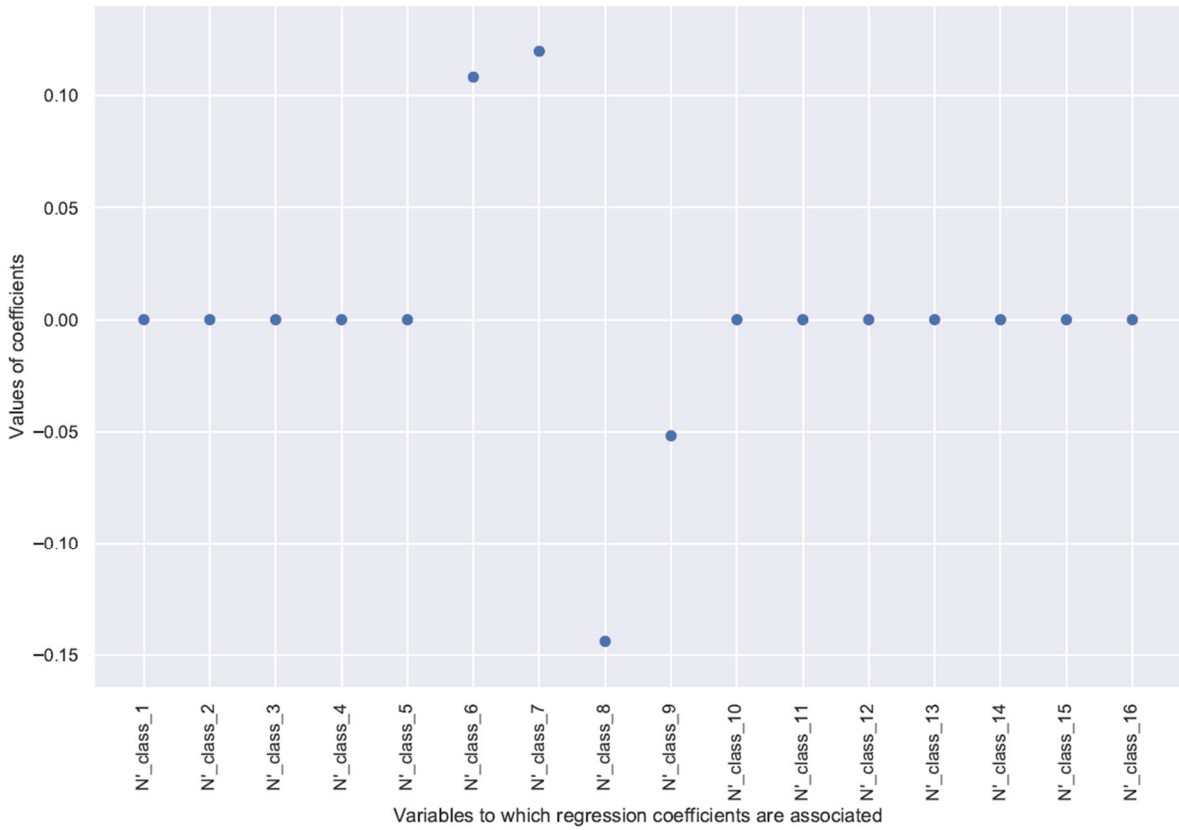

**Figure 5.** Regression coefficients for the model of diameter class 8, associated with transformed features $N_d'$.

The effects of the respective features are comparable within the same model despite different numbers of stems depending on the class (high for small classes to low for large classes) because these features have been scaled to the unit variance. The increment in the number of stems per ha $\Delta N_d$ of a given diameter class $d$ is positively influenced by the number of stems per ha in the two directly underlying classes $(d-1)$ and $(d-2)$. These are the trees whose growth will allow them to move from class $(d-1)$ to $d$ or from class $(d-2)$ to $d$ during the growth period.

The number of stems in class $d$, however, has a negative influence on the increment of class $d$. This effect corresponds to the trees that will leave class $d$ and move to the overlying classes during the growth period. The effect of the overlying classes is negative but more difficult to interpret. In general, only the directly overlying classes seem to have a slight negative influence on the increment of the number of stems in a given class. It can be assumed that there is some competition between close diameter classes, provided that there is some spatial proximity between trees of these classes. This can be the case in small and intermediate classes. It is more uncommon for the largest trees whose neighborhood at their height is generally more sparse (those trees have already been fostered by previous cuts). Classes much larger than a given diameter class did not seem to play a clear and strong role on the increment in that class, either because of a lack of strong competition between trees of different diameters (e.g., due to spatial differentiation) or because this competition effect is relatively constant across divisions and that without variability, the models fail to quantify this effect. For this reason, the corresponding features were not included in the model. The threshold for not including these features in the model, $d_{max}$ in Equation (7), depends on a given diameter class, hence the notation $d_{max}(d)$.

For class 1, the situation is different for the reasons already stated, which have to do with the lack of information on the underlying classes. In this case, the model performed better by keeping all features corresponding to the overlying classes. Figure 6 shows the value of the regression coefficients associated with features $N_d'$ (see Equation (7)) for the diameter class 1 model.

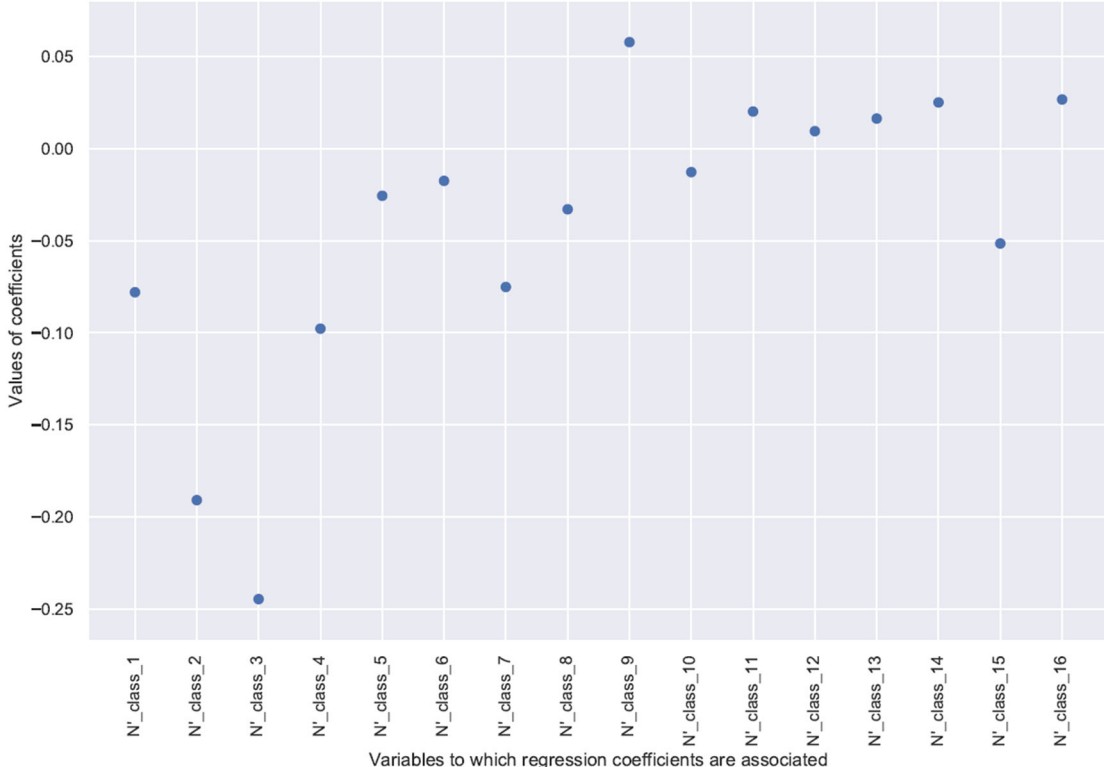

**Figure 6.** Regression coefficients for the model of diameter class 1, associated with transformed features $N_d'$.

The number of stems in class 1 has a negative effect on the increment in the same class. This effect is explained by the trees that will leave this diameter class during the growth period. The effect of the number of stems in the directly overlying diameter classes is also negative. This could be explained by the existing competition between the trees of these diameter classes and the trees of class 1. In contrast, the numbers of stems in the larger diameter classes do not appear to have a significant effect on the increment of class 1. The latter two findings could be explained by the hypotheses made above.

Figure 7 shows the regression coefficients associated with the feature $N'_{annual\ cut,d}$ (see Equation (7)) for the different models.

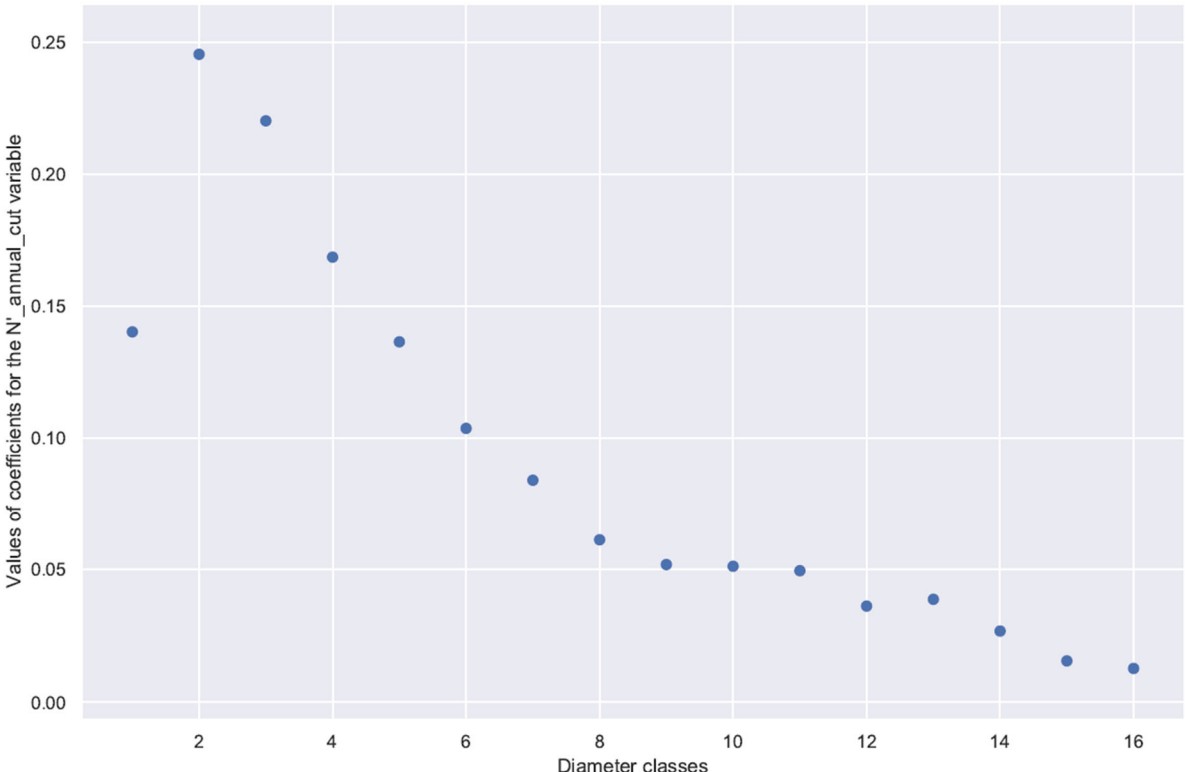

**Figure 7.** Regression coefficients corresponding to the transformed feature $N'_{annual\ cut,d}$.

Figure 7 shows the gross effect of cuts in a given diameter class on the increment in the number of stems in that same class, i.e., the indirect effect of cuts in a class on the growth of trees in that class but without the direct effect of tree removal. The indirect effect of cuts is positive for all diameter classes. Comparing diameter classes with each other is less obvious because increments (*a fortiori* transformed according to Yeo-Johnson) do not mean the same thing from one class to another; an increment in the number of stems will necessarily be lower in the larger diameter classes. However, it may be that intra-class competition is weaker in the larger diameter classes because the larger trees are more spatially separated from each other (they have been fostered by previous cuts), compared to the smaller classes. The pattern of regression coefficients associated with the feature $N'_{annual\ windfall,d}$ is similar to that in Figure 7.

### 3.2. Results of the Modeling with the Multilayer Perceptron

The best topology had two hidden layers with 10 neurons each (among those tested: one or two hidden layers, 5 to 100 neurons by layer in steps of five). The optimal value of the regularization parameter (L2) optimally defined by the gridsearch is 100 in both cases, an intermediate value among the values tested.

Even though a unique model is trained for all diameter classes, its performances are disaggregated by diameter class in order to compare the results with those of the linear models. Results are presented in Table 3.

**Table 3.** Comparison of performances for predicting $N_{after\ gross,d,i}$ between the linear models and the multilayer perceptron (MLP).

| Diameter Class | Diameter Range | Test R² Linear Models (Reminder) | Test R² MLP | Train R² MLP | Test RMSE MLP | Mean of Real $N_{after\ gross,d,i}$ Values |
|---|---|---|---|---|---|---|
| 1 | [ | 82.4% | 81.4% | 83.0% | 14.3 stems/ha | 74.9 stems/ha |
| 2 | [22.5;27.5[ | 93.5 | 89.7 | 91.5 | 8.3 | 59.1 |
| 3 | [27.5;32.5[ | 93.4 | 90.4 | 91.5 | 6.4 | 49.1 |
| 4 | [32.5;37.5[ | 92.9 | 85.8 | 88.2 | 5.6 | 39.1 |
| 5 | [37.5;42.5[ | 91.3 | 82.7 | 86.2 | 4.8 | 31.3 |
| 6 | [42.5;47.5[ | 89.2 | 81.1 | 84.1 | 4.0 | 24.0 |
| 7 | [47.5;52.5[ | 90.3 | 80.3 | 83.2 | 3.6 | 18.4 |
| 8 | [52.5;57.5[ | 88.7 | 78.8 | 81.2 | 3.1 | 13.3 |
| 9 | [57.5;62.5[ | 90.6 | 79.3 | 82.3 | 2.5 | 9.4 |
| 10 | [62.5;67.5[ | 90.4 | 75.5 | 80.5 | 2.1 | 6.1 |
| 11 | [67.5;72.5[ | 89.6 | 61.4 | 74.6 | 1.9 | 3.7 |
| 12 | [72.5;77.5[ | 88.0 | 47.0 | 66.7 | 1.4 | 2.1 |
| 13 | [77.5;82.5[ | 84.2 | 16.1 | 55.1 | 1.2 | 1.3 |
| 14 | [82.5;87.5[ | 74.5 | −85.5 | 20.3 | 1.0 | 0.6 |
| 15 | [87.5;92.5[ | 69.8 | −298.8 | −87.5 | 0.8 | 0.3 |
| 16 | [92.5;97.5[ | 65.9 | −1133.7 | −389.3 | 0.8 | 0.1 |
| All classes combined | | 90.3% | 96.1% | 96.5% | | |

The analysis of results by diameter class shows that the multilayer perceptron results in lower and more heterogeneous performance than the linear models. The performance is close for some diameter classes (the lower diameter classes for which a large volume of data are available), and it is inferior in the higher diameter classes. For classes 14, 15, and 16, for which there are limited data, the multilayer perceptron is apparently totally unable to predict future stem numbers. In the case presented in Table 3, the model with two hidden layers of 10 neurons each is relatively complex (in comparison to linear models presented in Section 3.1), which gives good results in the lower classes but produces overfitting and thus bad results in the higher classes. This hypothesis is confirmed by comparing the training and test R² at the level of the different diameter classes. These two scores are close for the low and intermediate diameter classes ($1 \leq d \leq 9$) but diverge for the higher diameter classes (and the higher the class). This reflects overfitting in the higher diameter classes. For the two highest classes (15 and 16), even the training R²s become negative, reflecting the fact that the MLP is no longer able to learn from the data even through overfitting. This is due to the scarcity of data in the higher classes. There are few trees this large in the data set and thus many null features for these diameter classes.

The performance of the multilayer perceptron seen on the whole data set is however better than that of the linear models. As the multilayer perceptron has been trained on the whole data set, its objective is to minimize the errors on the whole data set, which leads to this better global result.

The results of the permutation importance procedure are shown in Table 4.

**Table 4.** Results of a permutation importance procedure on the multilayer perceptron (with 30 repetitions). Only main features for which the loss in test $R^2$ is above 0.001 are mentioned in the table.

| Feature | Test $R^2$ Loss (Permutation Importance) |
|:---:|:---:|
| $N_d$ | 0.6522 |
| $d$ (diameter class) | 0.2012 |
| $G_d$ | 0.046 |
| $N_{overlying\ annual\ cut,d}$ | 0.0085 |
| $G_{cum,d}$ | 0.0026 |
| $G_{resinous,1}$ | 0.0025 |
| $N_{resinous,1}$ | 0.0021 |
| $N_{deciduous,1}$ | 0.0011 |

The information extracted by the multilayer perceptron is related to the initial state of the diameter class (above all the number of stems $N_d$ but also the basal area $G_d$ in this class), the diameter ($d$), an indicator of the competition regulation ($N_{overlying\ annual\ cut,d}$), an indicator of competition ($G_{cum,d}$), and information on the initial state of the 1st diameter class ($G_{resinous,1}$, $N_{resinous,1}$, $N_{deciduous,1}$).

*3.3. Results of Linear Modeling at the Forest Division Scale*

Linear models perform better for predictions at the diameter class level, which are important from a silvicultural point of view. In particular, linear models also produce good results for larger diameter classes. In addition, due to their simplicity and transparency, linear models are much easier to interpret in terms of forest growth, which facilitates the quality control. Linear models could be used as a decision support tool for the planning of cuts in forests under selective felling management.

The performance of these models is also convincing when tested in real situations. The linear models are applied on whole test divisions (completely unknown to the models) to predict the increment of the number of stems in each diameter class. These increments are then used to calculate the raw stem counts (i.e., without tree removal from cuts or windfalls) at the end of the growth period and can be compared to the actual values. Figures 8–11 show the results for four couples (division, inventory year).

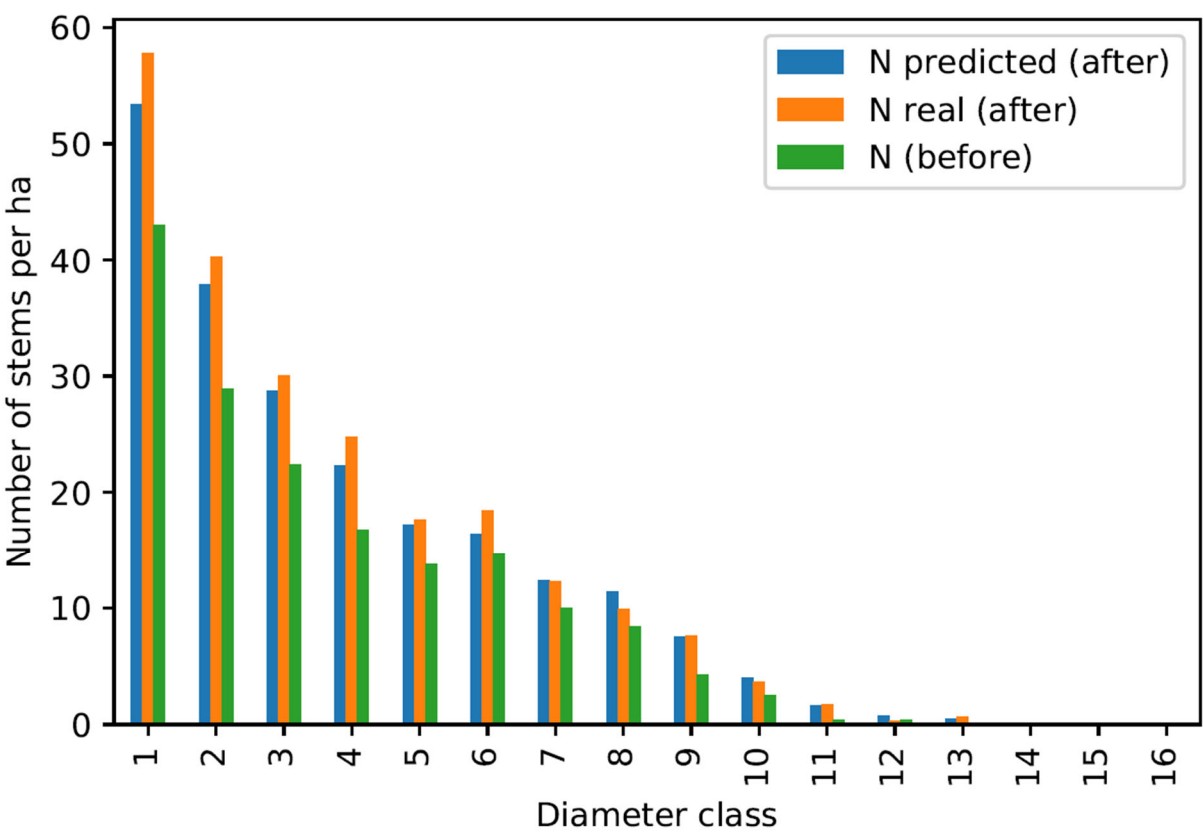

**Figure 8.** Overall prediction quality control for the division 1974 inventoried in the year 2003.

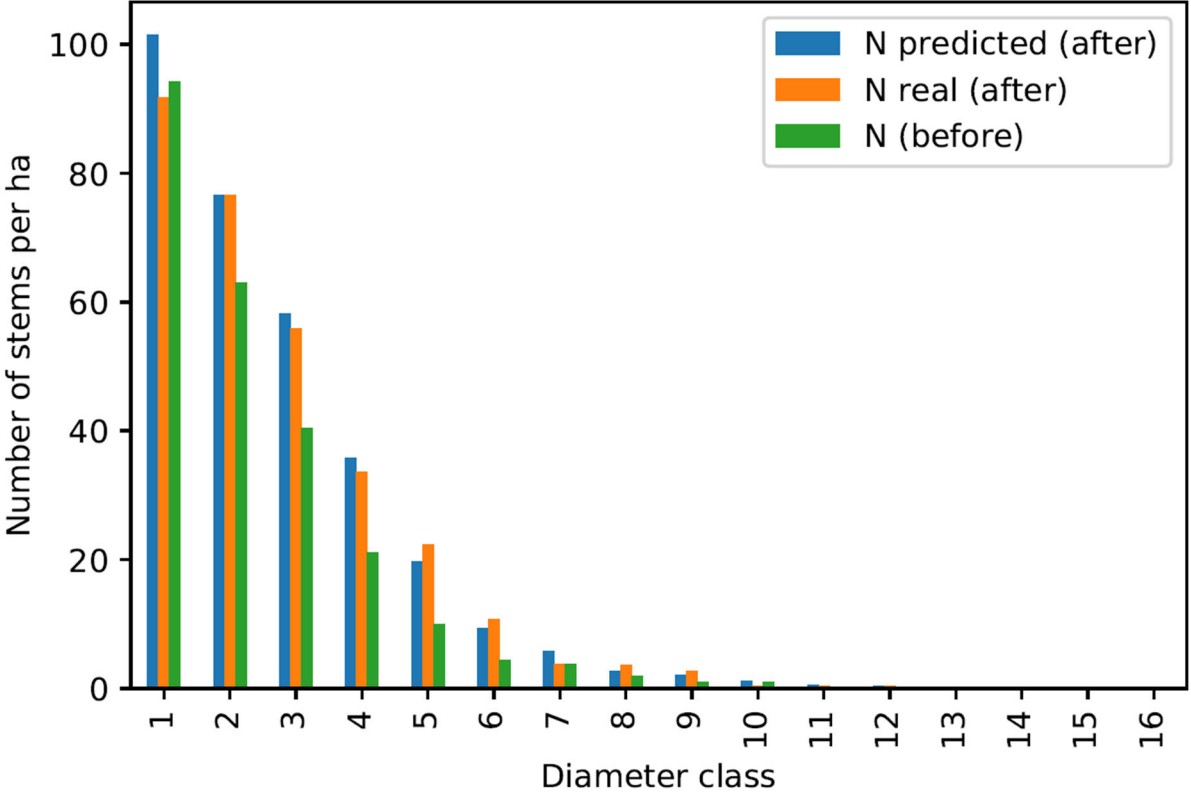

**Figure 9.** Overall prediction quality control for the division 2240 inventoried in the year 1947.

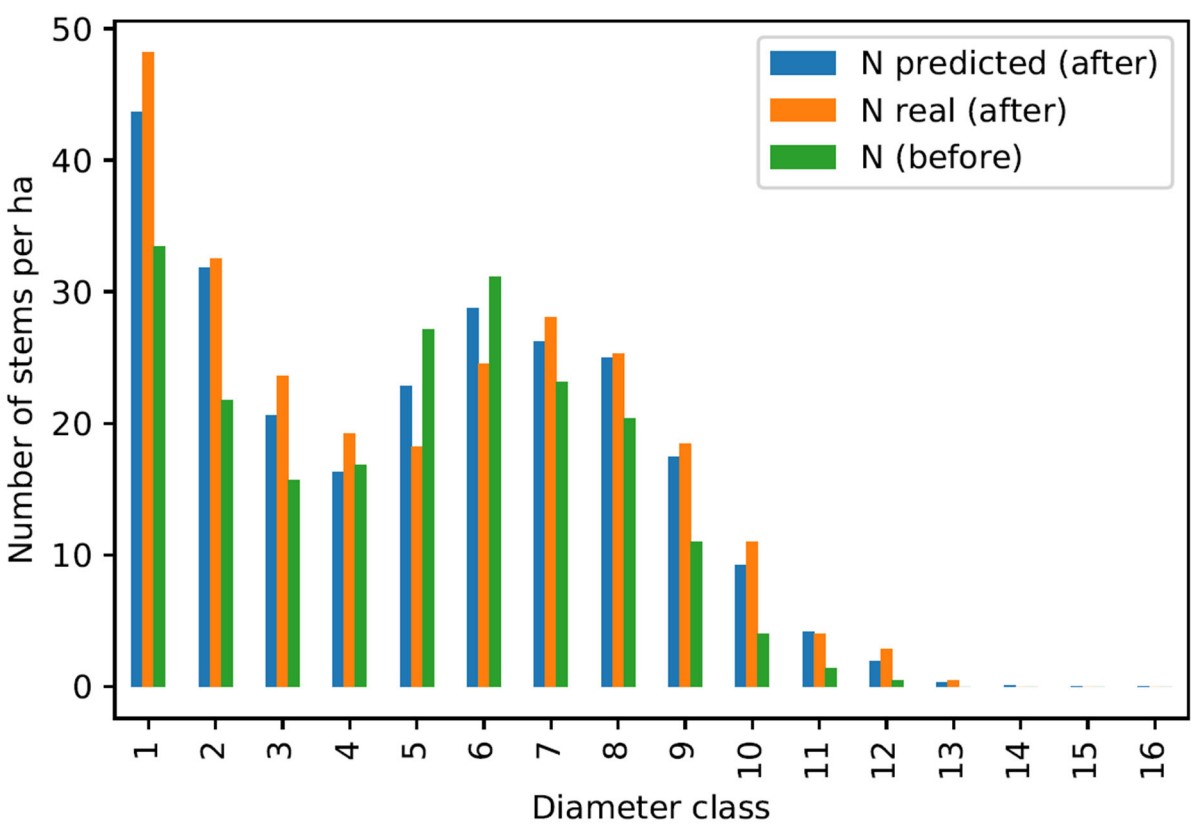

**Figure 10.** Overall prediction quality control for the division 809 inventoried in the year 1989.

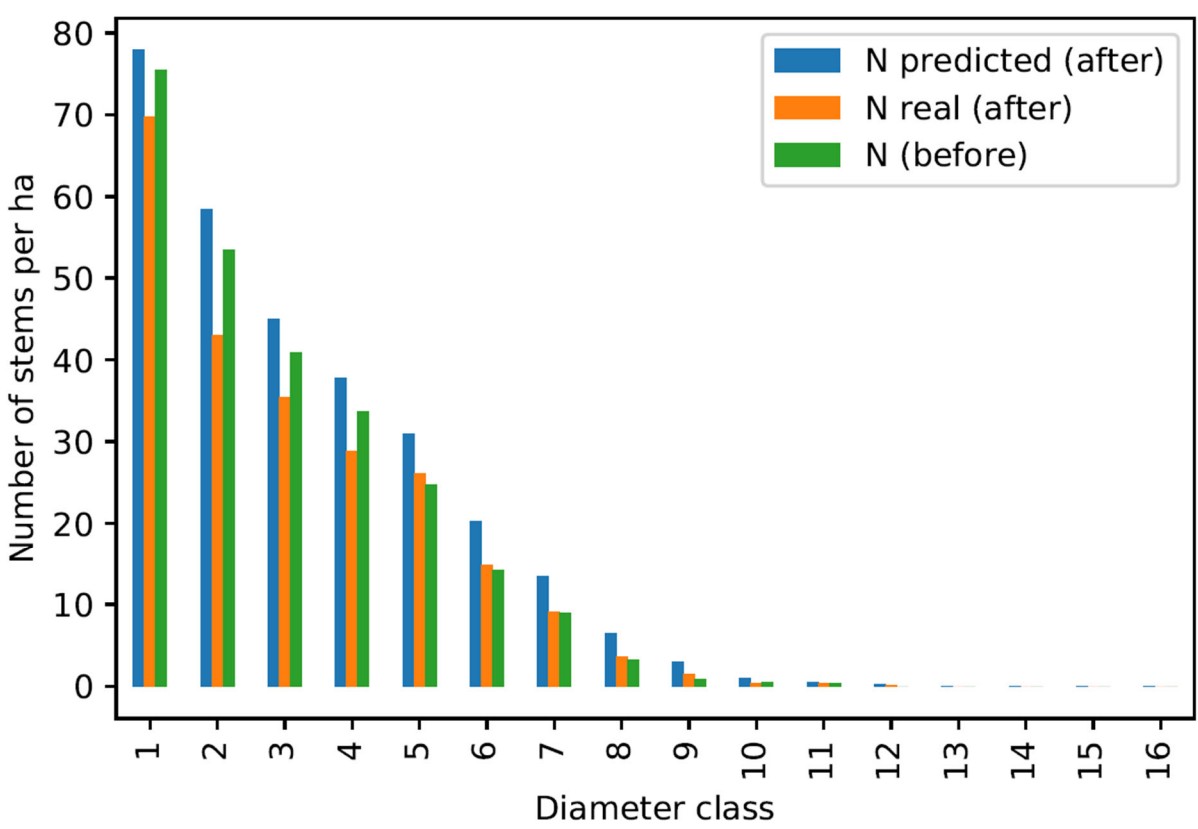

**Figure 11.** Overall prediction quality control for the division 1878 inventoried in the year 1983.

Figures 8–10 show quite contrasting situations for which the models produce good results. The predictions for the first diameter class are rather approximate due to the lower performance of this model. However, results are particularly convincing for intermediate classes, which correspond to the best performing models. Figure 11 shows an example where the models produce poor results, which happens for a small minority of the divisions tested.

Linear models also perform well for predicting increments of some aggregate features at the division level, e.g., number of stems per ha N, basal area per hectare G, and standing volume V. Prediction tests are this time performed on all couples (division, inventory year) to keep the number of points large. This is not a problem, as the overfitting of linear models is extremely limited.

For each couple (division, inventory year), the linear models are used to predict the $\Delta N$ and $\Delta G$ at the end of the corresponding growth period. The predicted values are then compared to the actual values. For $\Delta N$, an $R^2$ of 52.3% is obtained, or a mean absolute error of 13.2 stems.ha$^{-1}$. For $\Delta G$, an $R^2$ of 42.5% is obtained, or a mean absolute error of 1.37 m$^2$.ha$^{-1}$. Figures 12 and 13 show the corresponding residuals plots.

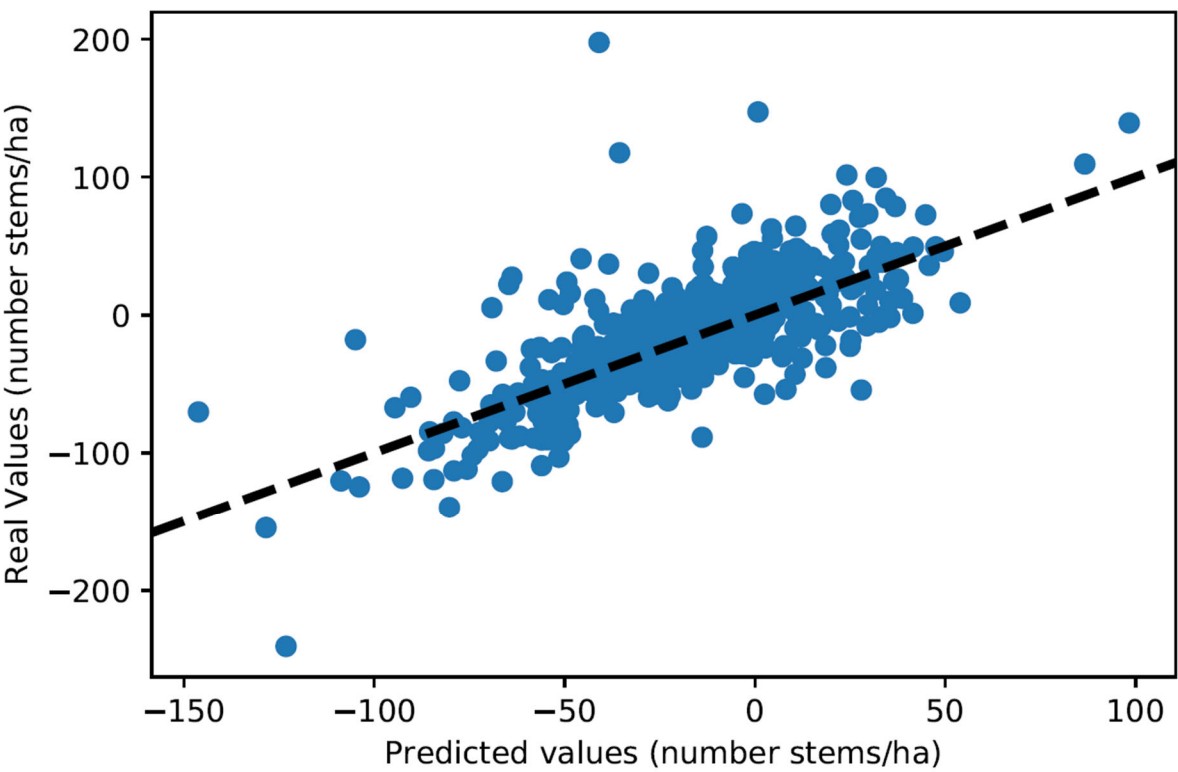

**Figure 12.** Residuals plot for $\Delta N$ predictions at the level of couples (division, inventory year).

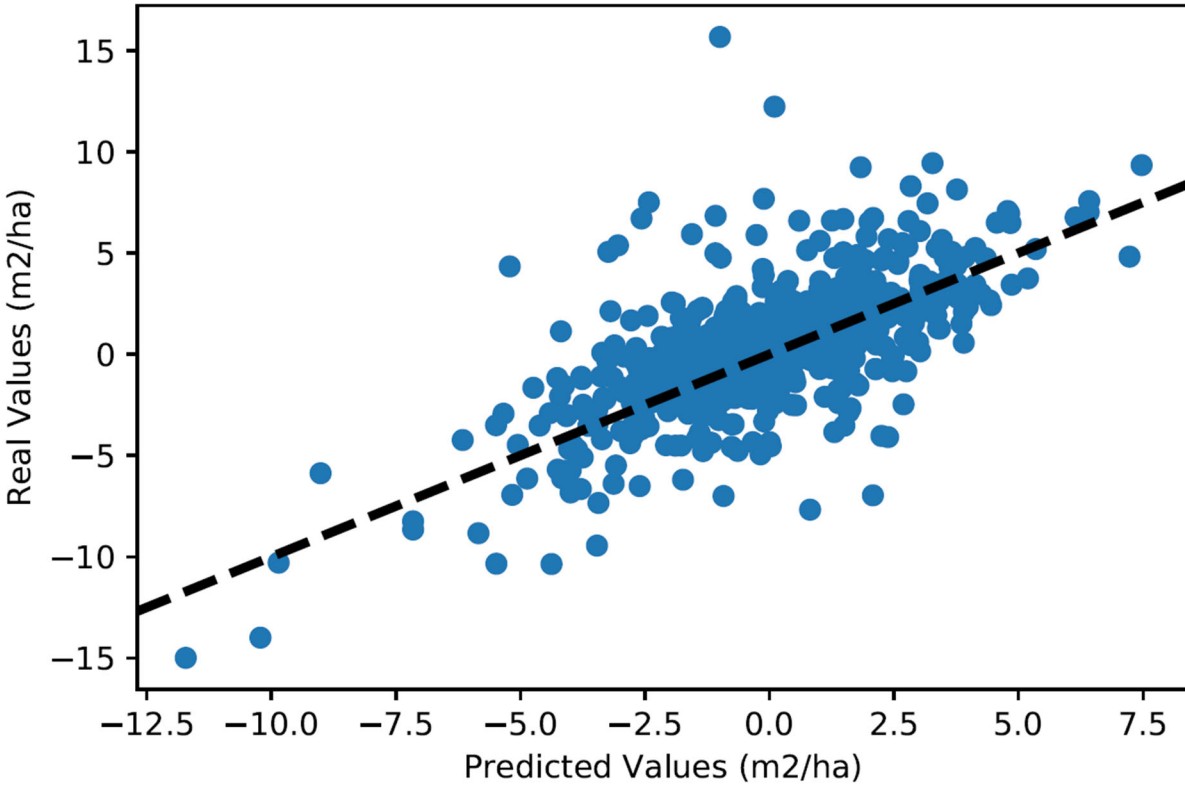

**Figure 13.** Residuals plot for $\Delta G$ predictions at the level of couples (division, inventory year).

In selective felling management practice, variables $N$, $G$ and $V$ are often disaggregated by large diameter classes; in the Canton of Neuchâtel, those classes are: small woods (17.5 to 32.5 cm), medium woods (32.5 to 52.5 cm) and large woods (>52.5 cm). Disaggregating the variables into these three classes provides concise information about the structure of a division. The predictive capacity of our models for $\Delta G$ in these three respective classes is tested. For small woods, we obtain an $R^2$ of 66.2%, i.e., a mean absolute error of 0.46 m².ha⁻¹. For medium woods, we obtain an $R^2$ of 78.9%, that is, a mean absolute error of 0.68 m².ha⁻¹. For large woods, we obtain an $R^2$ of 41.9%, that is, a mean absolute error of 0.73 m².ha⁻¹. Figures 14–16 show the corresponding residuals plots.

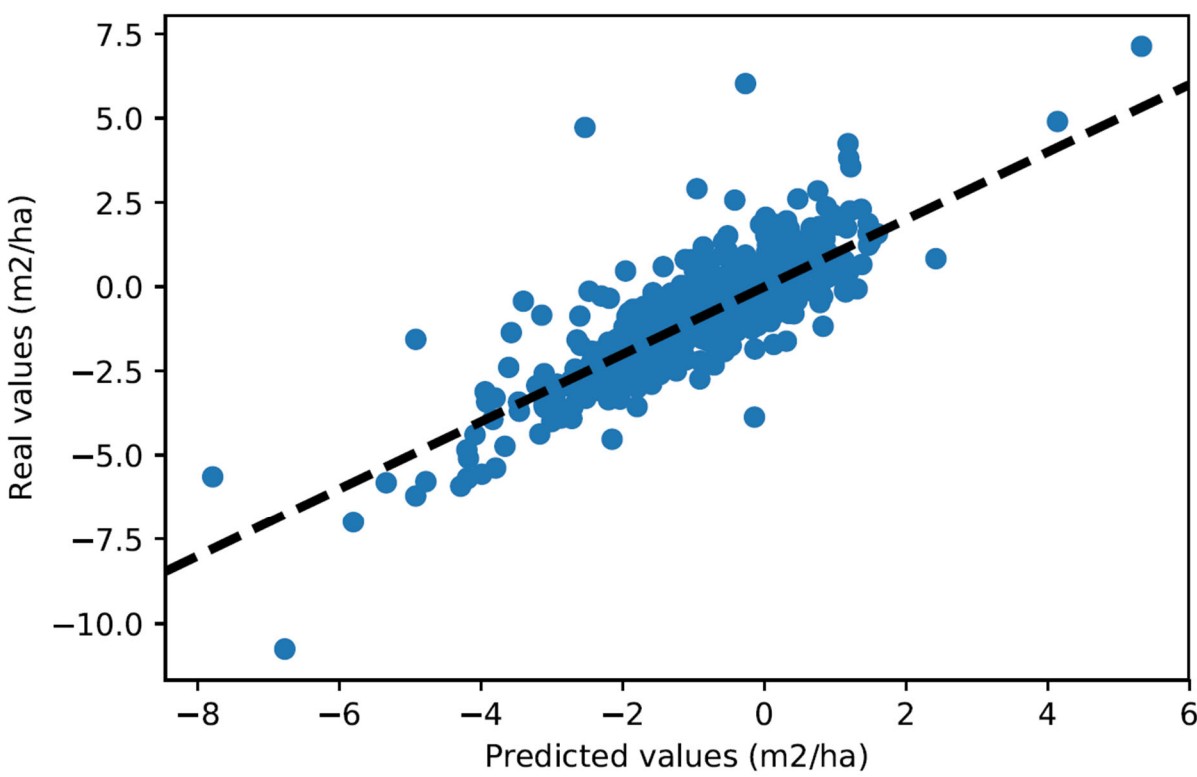

**Figure 14.** Residuals plot for $\Delta G$ predictions at the level of couples (division, inventory year) for the small woods class.

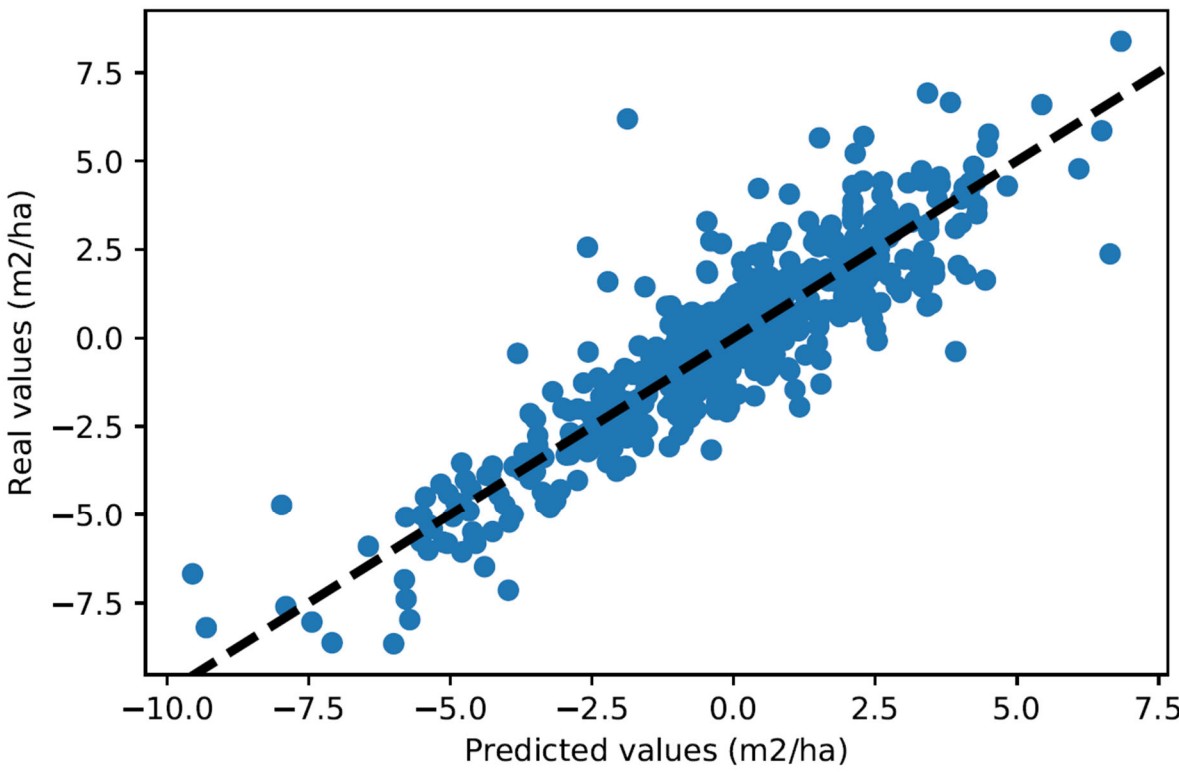

**Figure 15.** Residuals plot for $\Delta G$ predictions at the level of couples (division, inventory year) for the medium woods class.

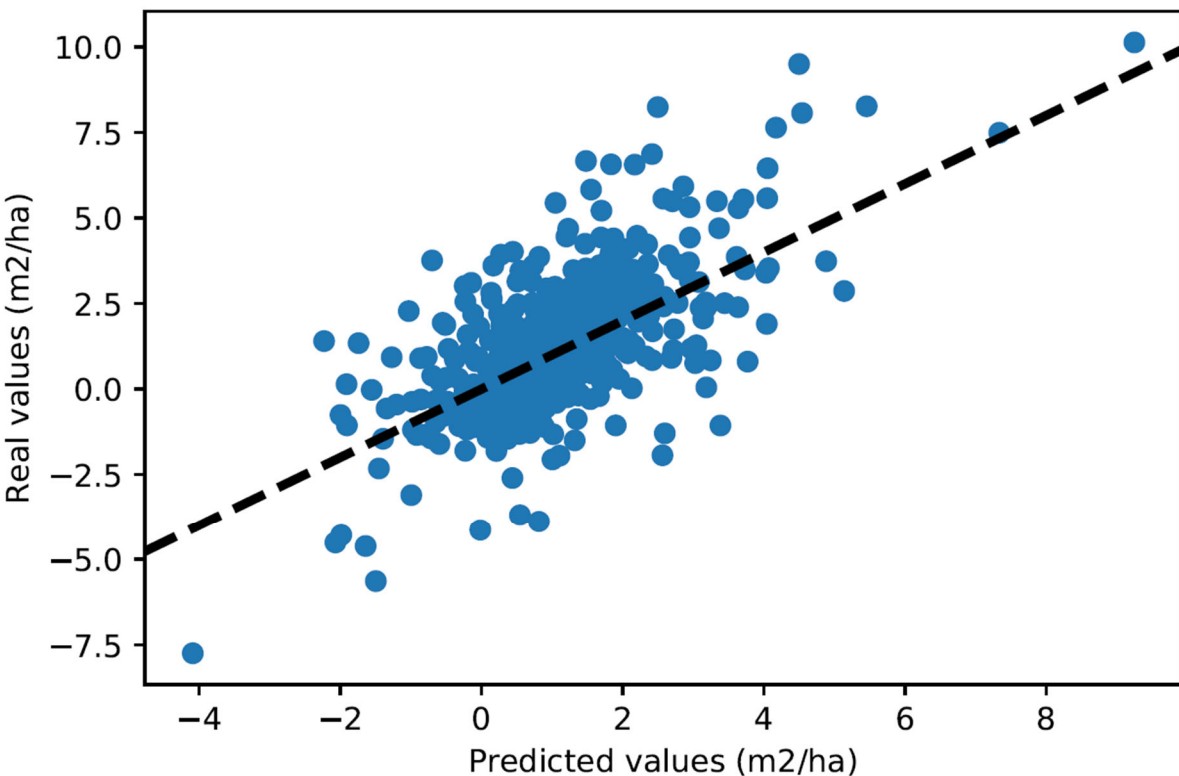

**Figure 16.** Residuals plot for $\Delta G$ predictions at the level of couples (division, inventory year) for the large woods class.

## 4. Discussion

### 4.1. Absolute and Comparative Performance of the Models Developed in this Study

The two modeling approaches followed in this study for modeling dynamics of the numbers of stems by diameter class both produce compelling results, which differ from each other. The linear models achieve better performance at the level of individual diameter classes. More specifically, performances of the two model types are close to each other for most diameter classes, but the performance of the linear models is much better for larger diameter classes.

For the linear models, test $R^2$ coefficients typically range from 50% to 70% (excluding class 1). These good performances are made possible by training a specific model for each diameter class (a form of data panelization). The absence of overfitting in the case of linear models indicates that the (large) volume of data available in this study is not limiting for those models. However, the comparatively poor performance of the model for diameter class 1 shows that the lack of data on the underlying classes ($d \leq 17.5\ cm$) is naturally limiting for predicting the evolution of the number of stems in this class.

As to the methodology, the implementation of a machine learning workflow, facilitated by existing technologies (in this case the scikit-learn library supported by the Python language), could significantly improve the performance of linear models by iteratively improving the features selection and transformation, as well as the selection of a loss function and a regressor. Train-test splits also allowed for an unbiased evaluation of model performances in terms of predictive capabilities, which was rarely proposed in past inferential statistical studies.

The multilayer perceptron provides the best performance at the global level for predicting the future number of stems with an average test $R^2$ of 96.1%. This is due to the fact that only one model was trained on the whole data set and not independently for each diameter class. The volume of data was considered insufficient to train a multilayer per-

ceptron per diameter class, as there would only be 580 observations per class. The multilayer perceptron shows heterogeneous performance in the diameter classes. The performances are close to the linear models for the lower and intermediate diameter classes (between 17.5 and 62.5 cm), but they decrease quickly toward the large diameters. The analysis of the difference between test and training $R^2$ shows an increasing overfitting for the diameter classes between 62.5 and 82.5 cm. Beyond 82.5 cm, the multilayer perceptron is unable to learn from the available data. It is therefore clear that the volume of data available for this study is limiting for modeling growth with a multilayer perceptron beyond 62.5 cm. From a methodological point of view, the implementation of the multilayer perceptron was much simpler than the fine-tuning of linear models.

### 4.2. Comparison of Performances with a Literature Benchmark

Both types of models developed in this study perform better than previously developed linear models for predicting dynamics at the diameter class level. For example, the model in [12] achieves an $R^2$ (on the whole data set) of 40% or less for predicting the probabilities of transition of trees between size classes. Our approach, differentiated by diameter class and based on a larger number of features (notably the initial diameter distribution) produces better results. It is not possible to directly compare our results with the results of models predicting individual tree diameter increment, but we can still note that the orders of magnitude of the test $R^2$ obtained are similar to ours, 57% for the best model in [4] and 53% for the best model in [9].

Aggregating the values predicted by our models for predicting increments of the number of stems per hectare, basal area per ha, or standing volume per ha at the division level produces relevant results for the practice of forest management. Nevertheless, the prediction performance is inferior to the performance of models specialized on this approach and based on machine learning methods. One model in [8] based on an artificial neural network achieves a test $R^2$ of 76%, and another model in [7], also based on an artificial neural network, achieves a test $R^2$ of 94% (versus 87% with linear regressions). This better performance is enabled by specialized models that directly predict the aggregate increment but probably also by the fact that these models are based on data from permanent plots.

### 4.3. Comparative Analysis of Feature Importance

The feature selection conducted in the case of linear models and the feature importance analysis conducted in the case of the multilayer perceptron result in partly different conclusions. It is interesting that the feature selection in the case of linear models has led without this being the initial goal to models based on the concept of passage rates from one diameter class to the next. Linear models based on passage rates are common in the literature [5,10–14], but they are usually specified as such initially. They are simple but general models and are consistent with forest growth theory. The mainly empirical feature selection conducted in this study shows that this is also a powerful and parsimonious type of model when the use of a linear model is imposed.

The analysis of feature importance in the case of the multilayer perceptron shows the high importance of a limited number of features. These results are consistent with the theory on forest growth and with the results of past studies. It is clear that the number of stems ($N_d$) or basal area ($G_d$) in a diameter class are good predictors of the number of stems in that same class in the near future. The role of diameter $d$ is also intuitive and has been demonstrated many times [5,10]. Most interesting is the important role played by indicators of competition (cumulative basal area $G_{cum,d}$) and of regulation of competition (cuts in overlying diameter classes $N_{overlying\ annual\ cut,d}$) between diameter classes. The role of $G_{cum}$ has already been demonstrated in previous studies [5,10] and is confirmed and quantified here in a flexible and general-value model. The role of cuts in the overlying diameter categories is clearly demonstrated here. The fact that the latter two variables ($G_{cum,d}$ and

$N_{overlying\ annual\ cut,d}$) were not equally important in the linear models shows that the corresponding effects are likely to be non-linear. However, the linear models perform well nonetheless by exploiting the information related to the initial structure at the division level, which also reflects the competitive conditions. Finally, the particular importance of the features related to the initial state of diameter class 1 ($G_{resinous,1}$, $N_{resinous,1}$, $N_{deciduous,1}$) is probably due to the difficulty of predicting recruitment in the absence of data on underlying classes, the initial state of this class 1 remaining the best predictor of its future state.

However, one should be careful not to draw too general conclusions about the importance of the features achieved in this study. For example, forest sites have a known influence on forest productivity but are not considered important in the models developed in this study. This does not imply that these parameters are negligible in general. The variability of forest sites is relatively small at the scale of the Neuchâtel forests managed according to selective felling. Furthermore, our models focus on short- to medium-term growth (about 10 years). On a larger spatial or temporal scale, forest sites have an important effect on forest productivity.

*4.4. Assessment of the Use of Data from the Canton of Neuchâtel*

Our results show that the inventory data from the Canton of Neuchâtel, collected through the *Méthode du contrôle*, generate valuable information to train convincing growth models at the diameter class level. The use of this type of data is an additional challenge compared to the use of data from permanent plots. Because increments are not tracked on the individual tree level, the calculation of increments is more approximate, for example, because of the lack of data on mortality. The lack of information on stems below 17.5 cm and of direct data on recruitment in the first diameter classes makes it difficult to model their dynamics. From this point of view, the performance of our models is limited. Another particularity of the data collected by the Canton of Neuchâtel compared to permanent plots is related to the inventory areas. The divisions used for this purpose are each about ten hectares in size instead of a few thousand square meters at most for the plots. Data at such a large scale are necessarily averaged, resulting in a probable reduction of variability in the data and reduced information for model training compared to the permanent plots.

The fact that the volume of data provided by the Canton of Neuchâtel was so large was clearly an added value for our study. The volume of data was clearly sufficient to train our linear models with a remarkable absence of overfitting. This large volume of data was also crucial for the training of the multilayer perceptron, which reached convincing performances. Nevertheless, the volume of data remained limiting for the latter, especially in the upper diameter classes.

## 5. Conclusions

The linear models developed in this study achieve sufficient performance for practical use. The predictions from these models are division specific in that the predicted dynamics depend on the initial diameter distribution and take into account competition between diameter classes to some extent. Linear models also have the advantage of being relatively transparent and therefore convincing for practitioners (coefficients can be directly interpreted by a professional knowledgeable in forest growth). Linear models, at least in this study, also predict fewer outliers than the multilayer perceptron.

Linear models can be used to predict the growth dynamics of a division over a period of 7 to 12 years. They can possibly be used recursively to predict dynamics over longer periods, but it is not recommended to simulate more than two or three growth periods, especially because the prediction of recruitment is limiting. The models can typically be used for better planning and implementation of the next cut in a given division. The model is not well suited to understanding the role of cuts on recruitment.

Before the model can be used in practice, it must first be tested under real conditions, for example on a current case study in the Canton of Neuchâtel.

*Implications and Management Options for Policy*

The (linear) models developed in this study could be used to assist in the planning of cuttings in uneven-aged stands. In their current state, they are applicable in stands managed according to the selection felling approach such as in the Canton of Neuchâtel, but after adaptation (i.e., re-estimation of the models on different sets of data), they could be applied to other uneven-aged management systems.

More specifically, the models could be used to:

- help plan the couple (harvest intensity, periodicity of interventions), specifically to a given forest stand (i.e., taking into account the current state of the stand);
- help in the silvicultural knowledge management, by serving as an analysis tool for training to selection felling (e.g., by simulating the potential effects of a cut on the dynamics of a stand);
- help concretely with selection felling in the field (for the selection of trees to be cut).

The possibilities of using such a tool would also be increased dramatically if it were possible to determine the initial structure of a stand in terms of diameter distribution from remote sensing data (especially LiDAR data). Data on the initial state of a stand are necessary for the application of the models, and remote sensing data are often available at lower cost and potentially more regularly than inventory data.

**Funding:** This study was financed by the Swiss Federal Office for the Environment (contract 00.0059.PZ/5933E7220). The author alone is responsible for the content of this article; the Federal Office for the Environment cannot be held responsible.

**Data Availability Statement:** The data used in this study were provided as a courtesy by the Canton of Neuchâtel (Switzerland). Some of the data are publicly available on the canton's geoshop (sitn.ne.ch/geoshop), free of charge when used for scientific purposes. Some data were not on the geoshop but were provided on motivated request and as a courtesy by the *Service de la faune, des forêts et de la nature* of the Canton of Neuchâtel.

**Acknowledgments:** I thank Christian Rosset from Bern University of Applied Sciences BFH for his feedback on the project design and acquisition.

**Conflicts of Interest:** The author declares no conflict of interest.

## Appendix A

**Table A1.** Average ranking of features through Recursive Feature Elimination (one feature is eliminated at each iteration until complete elimination of all features); these results are based on the optimal features transformation and the optimal regressor presented in Section 3.1.2; each average is calculated over feature ranks of the respective models for the 16 different diameter classes and over 10 train-test splits for each model (i.e., 160 values in total).

| Feature | Average Ranking (Recursive Feature Elimination) | Feature | Average Ranking (Recursive Feature Elimination) |
|---|---|---|---|
| $N_{annual\ cut,d}$ | 8.2 | $N_{resinous,15}$ | 43.0 |
| $N_d$ | 8.9 | $N_{deciduous,4}$ | 43.4 |
| $G_d$ | 9.6 | $G_{deciduous,2}$ | 43.5 |
| $G_{resinous,6}$ | 28.9 | $G_{resinous,15}$ | 43.6 |
| $N_{resinous,6}$ | 28.9 | $G_{deciduous,10}$ | 44.0 |
| $N_{annual\ windfall,d}$ | 29.1 | $N_{resinous,14}$ | 44.1 |
| $G_{resinous,10}$ | 30.6 | $G_{tot}$ | 44.8 |
| $N_{resinous,3}$ | 31.5 | $N_{deciduous,14}$ | 44.9 |

| | | | |
|---|---|---|---|
| $G_{resinous,5}$ | 32.3 | $N_{resinous,1}$ | 45.3 |
| $G_{resinous,3}$ | 32.4 | $N_{deciduous,9}$ | 46.0 |
| $N_{resinous,10}$ | 32.4 | $G_{deciduous,9}$ | 46.1 |
| $N_{resinous,5}$ | 32.5 | $N_{deciduous,10}$ | 46.2 |
| $G_{deciduous,8}$ | 32.9 | $G_{deciduous,14}$ | 46.6 |
| $N_{resinous,11}$ | 33.2 | $N_{deciduous,1}$ | 47.0 |
| $G_{resinous,11}$ | 33.4 | $N_{resinous,16}$ | 47.5 |
| $G_{resinous,7}$ | 33.5 | $G_{resinous,1}$ | 47.6 |
| $N_{deciduous,8}$ | 33.8 | $G_{deciduous,16}$ | 47.9 |
| $N_{overlying\ annual\ cut,d}$ | 34.9 | $N_{overlying\ annual\ windfall,d}$ | 48.0 |
| $G_{resinous,12}$ | 35.7 | Forest_site_13 | 48.1 |
| $N_{resinous,7}$ | 35.8 | $G_{deciduous,1}$ | 48.4 |
| Forest_site_8 | 36.0 | Forest_site_12 | 48.7 |
| $N_{deciduous,3}$ | 36.0 | $G_{resinous,16}$ | 49.5 |
| $N_{resinous,4}$ | 36.3 | Forest_site_2 | 49.6 |
| $G_{resinous,4}$ | 36.4 | $N_{tot}$ | 49.8 |
| $N_{resinous,12}$ | 38.5 | Forest_site_14 | 50.0 |
| $G_{resinous,9}$ | 38.5 | $N_{deciduous,5}$ | 50.0 |
| $G_{resinous,8}$ | 38.6 | $N_{deciduous,11}$ | 50.1 |
| $N_{resinous,9}$ | 38.9 | $G_{deciduous,5}$ | 50.3 |
| $N_{resinous,8}$ | 39.0 | $G_{deciduous,11}$ | 50.8 |
| $G_{resinous,13}$ | 39.1 | Forest_site_27 | 52.6 |
| $N_{resinous,13}$ | 39.3 | $G_{deciduous,15}$ | 53.1 |
| $G_{deciduous,3}$ | 39.6 | $N_{deciduous,16}$ | 53.3 |
| $N_{resinous,2}$ | 40.0 | $N_{deciduous,15}$ | 54.7 |
| $G_{deciduous,7}$ | 40.2 | Forest_site_11 | 54.9 |
| $N_{deciduous,7}$ | 40.5 | Forest_site_9 | 55.7 |
| $G_{resinous,2}$ | 40.9 | $N_{deciduous,12}$ | 55.9 |
| $G_{resinous,14}$ | 41.0 | $G_{deciduous,12}$ | 57.6 |
| $N_{deciduous,2}$ | 41.3 | Forest_site_17 | 57.8 |
| $N_{deciduous,6}$ | 41.6 | $N_{deciduous,13}$ | 58.3 |
| $G_{deciduous,6}$ | 42.0 | $G_{deciduous,13}$ | 59.4 |
| $G_{deciduous,4}$ | 42.6 | Forest_site_23 | 60.3 |
| $G_{cum.d}$ | 42.6 | | |

**Table A2.** Results of the Stepwise Regression with Backward Elimination; values are test $R^2$ for the respective models corresponding to the 16 different diameter classes; these results are based on the optimal features transformation and the optimal regressor presented in Section 3.1.2; the columns correspond to different sets of features, starting with all features in the left column and progressively and cumulatively eliminating or modifying groups of features going to the right; the bold column corresponds to the set of features selected for the final models in this study; all values are averaged on 10 different train-test splits.

| Diameter Class | All Features | then removing Forest Sites | then removing $N_{tot}/G_{tot}$ | then removing $G_{cum}$ | then removing $N_{overlying\ annual\ cut}$ and $N_{overlying\ annual\ windfall}$ | then removing G Features | then merging of Variables $N_{resinous}$ and $N_{deciduous}$ | then modifying of N Features | then removing $N_{annual\ cut}/N_{annual\ windf}$ |
|---|---|---|---|---|---|---|---|---|---|
| 1 | 12.4 | 13.2 | 12.9 | 14.6 | 12.5 | 13.5 | 11.9 | **11.9** | 10.6 |
| 2 | 50.1 | 53.5 | 53.3 | 53.3 | 52.8 | 51.6 | 53.3 | **54.1** | 51.5 |
| 3 | 61 | 63.9 | 63.7 | 63.6 | 62.7 | 62.3 | 63.5 | **64.1** | 61 |

| 4 | 65.9 | 67.8 | 68.7 | 68.7 | 69.3 | 67.3 | 65 | **65.9** | 63.7 |
|---|---|---|---|---|---|---|---|---|---|
| 5 | 65 | 68.4 | 68.8 | 68.7 | 69.2 | 67 | 64.8 | **65.2** | 61.9 |
| 6 | 61.2 | 63 | 64 | 64.1 | 64.2 | 62.4 | 59.4 | **60.1** | 57.1 |
| 7 | 60.1 | 60.7 | 60.3 | 60.1 | 60.7 | 56.9 | 56.2 | **54.6** | 50.2 |
| 8 | 50.9 | 53.9 | 53.2 | 53.1 | 53.6 | 50.7 | 46.6 | **48.7** | 44.8 |
| 9 | 48.9 | 48.8 | 50 | 47.6 | 49.3 | 47.8 | 46 | **48.7** | 42.6 |
| 10 | 54.7 | 54.7 | 54.3 | 54.9 | 55 | 53 | 50 | **51.9** | 46.8 |
| 11 | 48.2 | 48.7 | 48.8 | 47.9 | 49.8 | 49.1 | 49.8 | **52.5** | 45 |
| 12 | 50.7 | 51.2 | 51.8 | 50.1 | 50.6 | 52 | 51.3 | **53.1** | 47 |
| 13 | 56.9 | 56.5 | 57.8 | 59 | 58.4 | 58.4 | 59.2 | **61.5** | 56.4 |
| 14 | 55.1 | 52.5 | 55 | 53.6 | 56 | 56.9 | 57 | **57.6** | 41.5 |
| 15 | 51.3 | 49.1 | 50.1 | 50.8 | 51.8 | 53.2 | 54.9 | **56.4** | 46 |
| 16 | 48.2 | 50.4 | 49.4 | 50.9 | 51 | 52.8 | 53.9 | **49.9** | 28.5 |

**Appendix B**

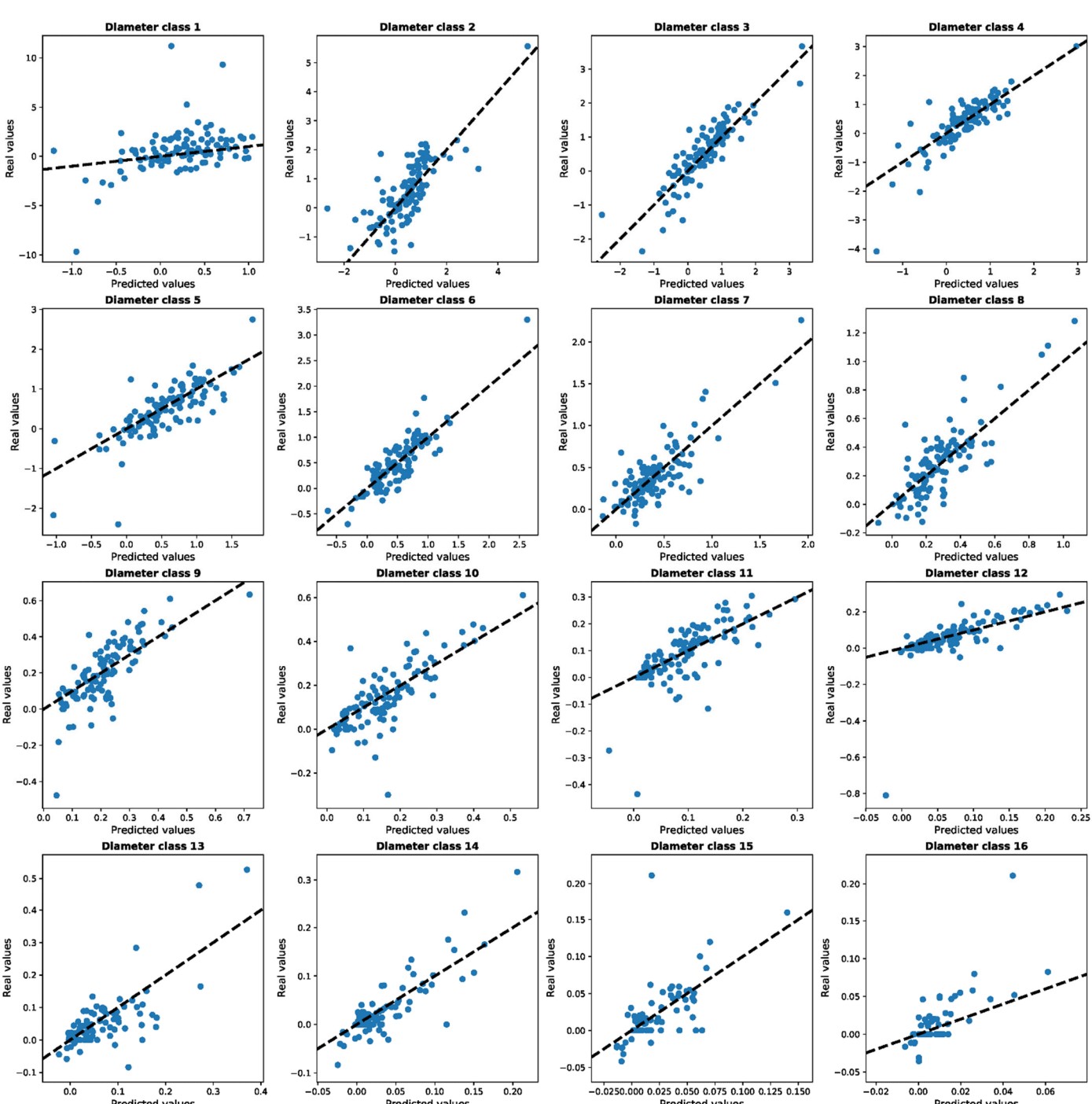

**Figure A1.** Residuals plots for linear models differentiated by diameter class presented in *Equation (7)* (classes from 1 to 16 where class center in cm = (#class − 1) × 5 + 20).

**Table A3.** Regression coefficients for linear models trained on the whole data set (class center in cm = (#class − 1) × 5 + 20).

| Model | Model Class 1 | Model Class 2 | Model Class 3 | Model Class 4 | Model Class 5 | Model Class 6 | Model Class 7 | Model Class 8 | Model Class 9 | Model class 10 | Model Class 11 | Model Class 12 | Model Class 13 | Model Class 14 | Model Class 15 | Model Class 16 |
|---|---|---|---|---|---|---|---|---|---|---|---|---|---|---|---|---|
| Diameter range | [17.5;22.5[ | [22.5;27.5[ | [27.5;32.5[ | [32.5;37.5[ | [37.5;42.5[ | [42.5;47.5[ | [47.5;52.5[ | [52.5;57.5[ | [57.5;62.5[ | [62.5;67.5[ | [67.5;72.5[ | [72.5;77.5[ | [77.5;82.5[ | [82.5;87.5[ | [87.5;92.5[ | [92.5;97.5[ |
| Intercept | 0.31567783 | 0.43808498 | 0.47340118 | 0.46051455 | 0.52286354 | 0.42056252 | 0.35215925 | 0.25705117 | 0.20267967 | 0.15179939 | 0.10445036 | 0.07182693 | 0.0590785 | 0.0288967 | 0.01538693 | 0.00806861 |
| N_class_1 | −0.08857224 | 0.95262586 | 0.30478168 | 0 | 0 | 0 | 0 | 0 | 0 | 0 | 0 | 0 | 0 | 0 | 0 | 0 |
| N_class_2 | −0.19883118 | −0.70449874 | 0.44469622 | 0.26800507 | 0 | 0 | 0 | 0 | 0 | 0 | 0 | 0 | 0 | 0 | 0 | 0 |
| N_class_3 | −0.2479029 | −0.31537605 | −0.47945268 | 0.33349209 | 0.19441854 | 0 | 0 | 0 | 0 | 0 | 0 | 0 | 0 | 0 | 0 | 0 |
| N_class_4 | −0.09295262 | −0.12675351 | −0.22661129 | −0.36086538 | 0.38562311 | 0.21856427 | 0 | 0 | 0 | 0 | 0 | 0 | 0 | 0 | 0 | 0 |
| N_class_5 | −0.03021211 | 0 | −0.10632899 | −0.21420664 | −0.44715699 | 0.19482813 | 0.14910786 | 0 | 0 | 0 | 0 | 0 | 0 | 0 | 0 | 0 |
| N_class_6 | −0.02551455 | 0 | 0 | −0.11447264 | −0.12177319 | −0.30391304 | 0.18257123 | 0.10812181 | 0 | 0 | 0 | 0 | 0 | 0 | 0 | 0 |
| N_class_7 | −0.09059655 | 0 | 0 | 0 | −0.05773986 | −0.1078003 | −0.24606661 | 0.11924328 | 0.06907012 | 0 | 0 | 0 | 0 | 0 | 0 | 0 |
| N_class_8 | −0.04804268 | 0 | 0 | 0 | 0 | 0.00040884 | −0.06378476 | −0.14591235 | 0.06992039 | 0.07125811 | 0 | 0 | 0 | 0 | 0 | 0 |
| N_class_9 | 0.03877847 | 0 | 0 | 0 | 0 | 0 | 0 | −0.05121676 | −0.09001726 | 0.06730688 | 0.04144019 | 0 | 0 | 0 | 0 | 0 |
| N_class_10 | −0.03040491 | 0 | 0 | 0 | 0 | 0 | 0 | 0 | −0.03252369 | −0.10502199 | 0.05030268 | 0.04166453 | 0 | 0 | 0 | 0 |
| N_class_11 | 0.00132175 | 0 | 0 | 0 | 0 | 0 | 0 | 0 | 0 | −0.06807874 | 0.04787912 | 0.02745242 | 0 | 0 | 0 | 0 |
| N_class_12 | −0.00334377 | 0 | 0 | 0 | 0 | 0 | 0 | 0 | 0 | 0 | −0.06532824 | 0.03742049 | 0.01877528 | 0 | 0 | 0 |
| N_class_13 | 0.01200967 | 0 | 0 | 0 | 0 | 0 | 0 | 0 | 0 | 0 | 0 | −0.00304633 | −0.03685599 | 0.02548971 | 0.01338672 | 0 |
| N_class_14 | 0.01124542 | 0 | 0 | 0 | 0 | 0 | 0 | 0 | 0 | 0 | 0 | 0 | −0.03028762 | 0.01327143 | 0.00454961 | |

| | | | | | | | | | | | | | | | | |
|---|---|---|---|---|---|---|---|---|---|---|---|---|---|---|---|---|
| N_class_15 | −0.06703446 | 0 | 0 | 0 | 0 | 0 | 0 | 0 | 0 | 0 | 0 | 0 | 0 | 0 | −0.01833805 | 0.0064068 |
| N_class_16 | 0.02649344 | 0 | 0 | 0 | 0 | 0 | 0 | 0 | 0 | 0 | 0 | 0 | 0 | 0 | 0 | −0.00812474 |
| N_annual_cut | 0.1378631 | 0.2531186 | 0.21802826 | 0.1655776 | 0.13114236 | 0.10657015 | 0.08537938 | 0.05979587 | 0.05113959 | 0.05160261 | 0.04996778 | 0.03751677 | 0.03917284 | 0.02788062 | 0.01534358 | 0.01277216 |
| N_annuel_windfall | 0.09950984 | 0.04107477 | 0.03255623 | 0.02292689 | 0.00990127 | 0.03090872 | 0.02697962 | 0.0147016 | 0.00961194 | 0.0032046 | 0.01048788 | 0.00560738 | 0.00773745 | 0.00713012 | 0.00603813 | 0.00204978 |

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
