# Peer review of "Exploring the Potential of Machine Learning for Modeling Growth Dynamics in an Uneven-Aged Forest at the Level of Diameter Classes: A Comparative Analysis of Two Modeling Approaches"

_forests, doi:10.3390/f13091432_

Round 1
Reviewer 1 Report
In this paper, the authors explore the use of machine learning methods for modeling growth dynamics in uneven-aged forests at the diameter class level, including fine-tuned linear models and an artificial neural network in Canton of Neuchâtel. However, several problems were identified during this review round.
1. Section 2.4.
As is well known, the feature engineering is of great importance in machine learning. The description of the feature engineering in this section is pretty simple, which might fail to convince the readers about why those features are selected while others are not. The authors are supposed to provide more information on the intuitive motivation and purposes of the selected features.
2. Section 2.5.
Although both the linear model and multilayer perceptron regressor are commonly used in recent researches, the author is still encouraged to provide more comprehensive descriptions in case there are readers who are not familiar with machine learning algorithms. For example, the author is supposed to provide several figures of linear model and multilayer perceptron regressor in this section.
3. Section 4.
Actually, I really appreciate the detailed presentation in Section3. However, the discussion needs drastic improvement.
a) The author simply owes the degradation of performance in the higher diameter classes (many 0-values in these data) due to an over-fitting, which is difficult to convince the readers. The author is supposed to provide more pieces of evidence before drawing a conclusion.
b) Considering the purpose of this manuscript is to explore the use of machine learning methods, the author is supposed to provide several ablation studies in this section. For example, the analysis of feature importance in exploited machine learning methods.
The reviewer also suggests that the writing should also be furtherly improved.
Reviewer 2 Report
This work explored the potential of machine learning for modeling forest growth dynamics, which is a valuable attempt promoting the development of forestry management technology in the era of big data. However, there is still much room for improvement in the statistical display and text expression of this article. Therefore, it is regrettable that I’ll give a suggestion of rejection this time, hoping the author can improve the paper and resubmit it.
1. I strongly suggest the author to shorten and condense all the expressions of the method and result parts, as they are currently too long and containing too much details, which will severely disturb the authors understanding. Many of these details can be removed or leave into the supplementary information.
2. Please show the information in Figure 3-5 as well as Figure 8-11 in a more comprehensive way. They are currently too simple and contain too little information.
3. Do not include methods and discussions in the results section, for example L486-491, L510-516, and L566-571.
4. Please provide more information for the forest density in the methods part.
5. If it is possible, please compare the model performance for different tree species.
6. Only using the coefficient of determination R2 to compare models seems to be too simple, please consider involving more indexes such as MSE, RMSE…
7. What is the radio between the training and testing datasets in your train-test-splits?
8. Please restate L481-483.
9. Please add more discussion to give your research in local scale a higher significance and implication for global scientists of forest managements.
Reviewer 3 Report
Paper is well written considering the assumptions f modelling with proper valiation technique.
Authros should provide one more extra section about "Implications and Managements options for policy". This will add essence to the studies.
Author Response
Dear reviewer,
I would like to thank you for your positive review and for your valuable feedback on my article.
I have now added a section called “Implications and Management options for policy” to my article. This section is concise and explains the uses that are really intended with the results of this study. This study is indeed part of an applied research project, which final goals are to generate added value to forest planning and management in practice.
Best regards and many thanks,
The author.
Round 2
Reviewer 1 Report
I have no suggestions for the manuscript in its present form.
Author Response
Dear reviewer, I thank you a lot for your previous feedbacks on my article.
Reviewer 2 Report
Thanks to the author for the edit and reply. I have no more suggestions for revisions, except that again, please fully simplify and beautify the text and graphics.
Author Response
Dear reviewer, I thank you a lot for your feedback on my article.